# Chronic intermittent hypoxia aggravated diabetic cardiomyopathy through LKB1/AMPK/Nrf2 signaling pathway

Bingbing Liu[1]☯, Jianchao Si[1]☯, Kerong Qi[1]☯, Dongli Li[1], Tingting Li[1], Yi Tang[1], Ensheng Ji[1,2]*, Shengchang Yang[1,2]*

**1** Department of Physiology, Hebei University of Chinese Medicine, Shijiazhuang, Hebei, People's Republic of China, **2** Hebei Technology Innovation Center of TCM Combined Hydrogen Medicine, Shijiazhuang, Hebei, People's Republic of China

☯ These authors contributed equally to this work.

* yscdekaoyan@163.com (SY); jesphy@126.com (EJ)

**Data Availability Statement:** All relevant data are within the manuscript and its Supporting Information files.

**Funding:** This work was supported by the National Natural Science Foundation of China (82274617),

## Abstract

Chronic intermittent hypoxia (CIH) may play an important role in the development of diabetic cardiomyopathy (DCM). However, the exact mechanism of CIH-induced myocardial injury in DCM remains unclear. In vivo, the db/db mice exposed to CIH were established, and in vitro, the H9C2 cells were exposed to high glucose (HG) combined with intermittent hypoxia (IH). The body weight (BW), fasting blood glucose (FBG) and food intake were measured every two weeks. The glycolipid metabolism was assessed with the oral glucose tolerance test (OGTT) and insulin resistance (IR). Cardiac function was detected by echocardiography. Cardiac pathology was detected by HE staining, Masson staining, and transmission electron microscopy. The level of reactive oxygen species (ROS) in myocardial tissue was detected by dihydroethidium (DHE). The apoptosis was detected by TUNEL staining. The cell viability, ROS, and the mitochondrial membrane potential were detected by the cell counting kit-8 (CCK-8) assay and related kits. Western blotting was used to analyze the liver kinase B1/AMP-activated protein kinase/ nuclear factor-erythroid 2-related factor 2 (LKB1/AMPK/Nrf2) signaling pathway. CIH exposure accelerated glycolipid metabolism disorders and cardiac injury, and increased the level of cardiac oxidative stress and the number of positive apoptotic cells in db/db mice. IH and HG decreased the cell viability and the level of mitochondrial membrane potential, and increased ROS expression in H9C2 cells. These findings indicate that CIH exposure promotes glycolipid metabolism disorders and myocardial apoptosis, aggravating myocardial injury via the LKB1/AMPK/Nrf2 pathway in vitro and in vivo.

## Introduction

Diabetic cardiomyopathy (DCM) is a representative complication of type 2 diabetes mellitus (T2DM), which is mainly manifested as systolic and diastolic dysfunction along with cardiac fibrosis, glycolipid metabolism disorders, and elevated oxidative stress [1, 2]. Recent studies

the Hebei Natural Science Foundation (H2022423352, H2022423370), the Science and Technology Project of Hebei Education Department (QN2023159) and the Medical Science Research Projects of Health Commission of Hebei Province (20231573). Author Contributions: Conceptualization: Shengchang Yang, Ensheng Ji. Data curation: Bingbing Liu, Jianchao Si, Shengchang Yang, Ensheng Ji. Formal analysis: Dongli Li, Tingting Li, Yi Tang, Shengchang Yang, Ensheng Ji. Funding acquisition: Shengchang Yang, Ensheng Ji. Investigation: Dongli Li, Tingting Li, Yi Tang. Methodology: Bingbing Liu. Project administration: Shengchang Yang, Ensheng Ji. Software: Kerong Qi. Visualization: Jianchao Si. Writing – original draft: Bingbing Liu, Jianchao Si, Kerong Qi. Writing – review & editing: Shengchang Yang, Ensheng Ji.

**Competing interests:** The authors have declared that no competing interests exist.

have shown that sleep-disordered breathing, especially obstructive sleep apnea (OSA), is closely related to glycolipid metabolism disorders and T2DM [3]. OSA is independently related to metabolic syndrome and insulin resistance (IR), which is strongly associated with the risk of adverse cardiovascular events [4]. OSA can lead to left ventricular diastolic and systolic dysfunction [5], which is associated with DCM [6]. Therefore, we explored the role of OSA in the pathogenesis of DCM, aiming to provide potential prevention and treatment approaches for DCM patients with OSA.

Chronic intermittent hypoxia (CIH) is a typical pathophysiological feature of OSA, which is associated with T2DM and cardiovascular disease [7, 8]. Badran et al. [9] found that CIH could aggravate the level of oxidative stress in intermittent hypoxia diabetic mice, further impairing endothelial dysfunction of db/db mice. Through the model of chronic intermittent hypoxia combined with the diabetes mellitus, Peng and Hu [10] found that CIH aggravated glycolipid metabolism disorders and the level of oxidative stress in the liver and kidney of diabetic rats, further aggravating the damage of the liver and kidney. At present, it is unclear whether CIH could be a factor in accelerating the progression of DCM.

Oxidative stress plays a crucial role in the pathogenesis of DCM [11]. The nuclear factor-erythroid 2-related factor 2 (Nrf2) pathway is the most important endogenous antioxidant stress pathway, which has a positive effect on resisting oxidative damage and the inflammatory response in DCM [12]. The AMP-activated protein kinase (AMPK)/Nrf2 pathway is related to antioxidant stress damage to the heart and brain. The downstream product of the AMPK/Nrf2 pathway, heme oxygenase-1 (HO-1), is a classic antioxidant enzyme and has a good inhibitory effect on oxidative stress and inflammation [13, 14]. Liver kinase B1 (LKB1), as the main upstream protein kinase of AMPK, promotes the phosphorylation and activation of AMPK, which is associated with the growth and metabolism of cells. The LKB1/AMPK signal transduction pathway plays an important role in glycolipid metabolism [15]. Related studies have shown that CIH inhibits the expression of the LKB1/AMPK pathway [16, 17]. In addition, CIH can induce ROS production and disrupt the oxidative/antioxidant balance of cells [18], and excessive ROS can inhibit AMPK inactivation by inhibiting the phosphorylation of LKB1 [19]. Therefore, we speculated that the LKB1/AMPK/Nrf2 signaling pathway may play a key role in the progression of CIH exacerbating DCM.

## Materials and methods

### Animals and treatment

Male diabetic db/db mice and their non-diabetic db/m mice (4-5weeks old) were purchased from Changzhou Cavens Laboratory Animal Co., Ltd. (License number: SCXK(Su)2016-0010). Mice were housed in a standard environment which was maintained on a 12 h light/dark cycle at constant room temperature (22±2˚C) with free access to food and water. All procedures involving animal and experimental protocols were approved by the Animal Care and Use Committee of Medical Ethics of Hebei University of Chinese Medicine (on Mar. 21, 2023; Permit No. DWLL202203123).

The db/db mice (n = 16) were randomly divided into the db/db group and CIH group. The db/m mice were used as the control group (n = 8). The treatment of CIH was started as previously described [9]. Briefly, db/db mice were placed in special cages with a controlled gas delivery system that regulated the flow of air, nitrogen, and oxygen into the cages. The concentration of oxygen in the chamber for the CIH group was changed from 21% to 5%, 8 h/day for 8 weeks. The db/m and db/db group mice were kept under air-conditioning. After exposure to CIH for 8 weeks, all mice were euthanized by cervical dislocation after intraperitoneal injection of 30 mg/kg of sodium pentobarbital. Then, the cardiac tissue was isolated.

## The measurement of biochemical parameters

The levels of total cholesterol (TC), triglyceride (TG) and low-density lipoprotein cholesterol (LDL-C) in mice were detected by an automatic biochemical analyzer (Changchun Huili Bio-tech Co., Ltd., Changchun, China). The myocardial tissue was collected to prepare myocardial homogenate, and the myocardial superoxide dismutase (SOD), glutathione peroxidase (GSH-Px), and malondialdehyde (MDA) levels in each group of mice were detected by using related kits (Nanjing Jiancheng Bioengineering Institute, Nanjing, China). The levels of creatine kinase myocardial band (CK-MB), lactate dehydrogenase (LDH) and cardiac troponin I (cTnI) were detected by automatic blood biochemical detector.

## Insulin resistance (IR) and Oral glucose tolerance test (OGTT)

The insulin level was detected by using a mouse insulin ELISA (abcam, UK). The following formula was used to calculate IR according to the following formula: Homeostatic Model Assessment for IR (HOMA-IR) = fasting insulin ($\mu$IU/L) × fasting glucose (mmol/L)/22.5. For the OGTT test, all mice were fasted for 15 h and given 2 g/kg of 20% glucose solution after FBG measurement. Blood samples were taken from tail veins at 15, 30, 60, and 120 min after gavage to measure the blood glucose level. The glucose-time curve was plotted and the area under the curve (AUC 0–120 min) was calculated.

## Echocardiography

The mice were anesthetized by inhalation with isoflurane (2%) and fixed in the supine position after full anesthesia. B-mode is the basic imaging mode of ultrasonic imaging, in which images of the anatomical structure of animals are used to locate the long and short axes of mice. M-mode was used to measure the left ventricular end diastolic diameter (LVDd), left ventricular end systolic diameter (LVDs), left ventricular fractional shortening (LVFS), and left ventricular ejection fraction (LVEF). Color Doppler-mode was used to measure the velocity ratio of the E peak to the A peak in the cardiac mitral valve (E/A).

## Histological examination

The heart tissue was fixed in 4% paraformaldehyde for 24 h for soaking, fixation, and dehydration, and then the wax-soaked tissue was embedded in an embedding machine. The sections were sliced on a paraffin slicer at a slice thickness of 4$\mu$m. The sections were stained in a dye solution and the morphological changes of the heart tissue were observed under the microscope.

Tissue blocks of about 1 mm$^3$ in size were removed from the apex of the heart and fixed in 2.5% glutaraldehyde (pH 7.4). Ultrathin sections were made and the ultrastructure was observed by electron microscopy.

The heart was fixed on ice in 4% paraformaldehyde for 2 h, and then transferred to a 30% sucrose solution for dehydration at 4˚C overnight, embedded with OCT, sliced with a frozen microtome with a thickness at 5$\mu$m, and fixed in cold acetone for 10min.

## Measurement of GLUT4 and Nrf2 expression

The frozen sections of the myocardium of mice were a sealed with 5% BSA blocking buffer for 1h at room temperature, GLUT4 (1:100, 66846-1-Ig, Proteintech, China) and Nrf2 (1:100, A0674, BOSTER, China) antibody was diluted and dropped onto the sections, and incubated at room temperature for 2 h. PBS was washed, and the diluted fluorescently labeled secondary antibody was added for 1h at 37˚C for protection from light, and the nuclear was stained with

diluted DAPI. The film was sealed and placed under confocal microscope to observe and photograph.

### TUNEL staining

The heart tissue embedded in paraffin was sectioned, dewaxed in xylene solution, hydrated with ethanol from a high to low concentration, washed three times with distilled water with phosphoric acid buffer, and incubated with TUNEL reagent at 37˚C for 1h, observed under a fluorescence microscope.

### Dihydroethidium staining

The myocardial tissues of mice in each group were taken for frozen sections, and incubated in 10μmol/L DHE solution for 90 min under dark conditions at 37˚C, washed with PBS, and photographed under fluorescence microscope after sealing. The fluorescence absorbance value was calculated as the level of intracellular ROS.

### Cell culture and treatment

The H9C2 cells were purchased from Procell Life Science & Technology Co., Ltd (Cat. No.: CL-0089; Wuhan, China). The Metformin was purchased from Shanghai yuanye Bio-Technology Co., Ltd (Cat. No.: B25331; Shanghai, China). The A-769662 was purchased from MedChemExpress LLC (Cat. No.: 844499-71-4; Shanghai, China). The H9C2 cells were divided into three groups: control (CON) group, high glucose (HG) group, and IH+HG group. The H9C2 cells were seeded in 24-well plates. The H9C2 cells were used to induce DCM in vitro under the HG condition. We cultured the H9C2 cells with 30mM of HG for 48 h. The IH+HG group was cultured in the hypoxic chamber of IH cells, and the oxygen concentration in the hypoxic chamber was cycled between 21%-1%, 10 min/cycle, and cultured for 48h.

### Cell viability

The H9C2 cells were prepared into a cell suspension and inoculated into 96-well plates at density of $5 \times 10^4$ cells/well. The Cell Counting Kit-8 (C0037, Beijing, China) was used to detect cell viability, CCK-8 solution (10μL) was added to each well, and the cells were incubated at 37˚C for 1 h. The absorbance value of each well was measured at 450 nm with a microplate reader, and the cell activity was calculated.

### Measurement of ROS production

The H9C2 cells were seeded in 24-well culture plates at a density of $5 \times 10^4$ cells in each well and incubated for 24 hours before treatment. Assay for ROS Assay Kit (S0033S, Beyotime, China), dilute DCFH-DA with serum-free DMEM medium at the ratio of 1:1000, add diluted with DCFH-DA 200 μL to each well, the cells were incubated in 5%$CO_2$ and 37˚C for 20 min and washed three times with serum-free cell culture medium. Finally, the cells were observed and photographed under an inverted fluorescence microscope and analyzed with Image J software.

### Measurement of mitochondrial membrane potential

The H9C2 cells were seeded in 24-well culture plates at a density of $5 \times 10^4$ cells in each well and incubated for 24 h before treatment. Mitochondrial Membrane Potential and Apoptosis Detection Kit with Mito-tracker Red CMXRos and Annexin V-FITC (C1071M, Beyotime, China) was used to detect the mitochondrial membrane potential. The medium and JC-1

working solution were added in equal proportions, fully mixed, and incubated at 37˚C for 30 min. After incubation, the liquid was removed and the cells were washed three times. Finally, the cell membrane potential was observed and photographed under an inverted fluorescence microscope.

## Western blotting

The protein concentration was quantified with the BCA Protein Assay Kit. Then 4× protein gel electrophoresis loading buffer was added to the protein samples, which were denatured at 100˚C for 5 min. The SDS-PAGE gel was prepared, samples were added for electrophoresis, and electricity was transferred to PVDF membrane. Seal with 5% skim milk powder sealing solution for 2 h, adding primary antibody and incubating overnight at 4˚C: Tubulin (1:10000, GTX628802) and Caspase-3(1:1000, #14220) from CST; p-AMPK (1:2000, AP0116) and HO-1 (1:1000, A1346) from ABclonal; PI3K (1:1000, AF6241) from Affinity; LKB1 (1:1000, 10746-1-AP), p-LKB1 (1:5000, 80127-1-RR), AMPK (1:500, 10929-2-AP), p-AKT (1:5000, 66444-1-Ig), AKT (1:5000, 60203-2-Ig), GLUT4 (1:3000, 66846-1-Ig) from Proteintech; Bax (1:1000, GB12690) from Seville; Bcl-2 (1:1000, BA0412) and Nrf2 (1:1000, A0674) from BOSTER; Lamin-B1(1:1000, SI17-06) from Huabio. After washing, the membrane was incubated for 1 h with second antibody (1:8000, Servicebio, China) and the proteins were detected with ECL luminescent solution. The gray value was analyzed by Image J software.

## Statistical analysis

Data are expressed as means ± S.E.M. The results were analyzed by SPSS 21.0 software. One-way ANOVA was used for intergroup mean comparison. The $p < 0.05$ was considered to be statistically significant.

## Results

### Effects of CIH exposure on the metabolic characteristics of db/db mice

As shown in Fig 1A and 1B, compared to the db/m group, the levels of BW, FBG and food intake in the db/db group and the CIH group were increased ($p < 0.01$). Compared to the db/db group, the BW of the CIH group was decreased ($p < 0.05$). There was no statistical significance in FBG between the db/db group and the CIH group, but the fluctuation of FBG in the CIH group was increased. Compared to the db/m group, the food intake of db/db group and CIH group was significantly increased ($p < 0.01$, Fig 1C). Compared to the db/db group, the food intake of the CIH group was decreased ($p < 0.05$). The changes in lipid metabolism in the db/db mice are shown in Fig 1D–1F. Compared to the db/m group, the levels of TC, TG, and LDL-C in the db/db group and CIH group were increased ($p < 0.05$). Compared to the db/db group, the levels of TC, TG, and LDL-C in the CIH group were increased ($p < 0.05$). As shown in Fig 1G and 1H, after 0–30 min of intragastric administration of glucose solution, the level of blood glucose in each group was increased; and after lavage for 30–120 min, the level of blood glucose was gradually decreased. Compared to the db/m group, the level of blood glucose in the db/db group and CIH group was significantly increased ($p < 0.01$). Compared to the db/db group, the level of blood glucose in the CIH group was increased after 15min of intragastric glucose solution ($p < 0.05$). Compared to the db/m group, the AUC value of the OGTT in the db/db group and CIH group was significantly increased ($p < 0.01$). Compared to the db/db group, the AUC value of the OGTT in the CIH group was increased ($p < 0.05$).

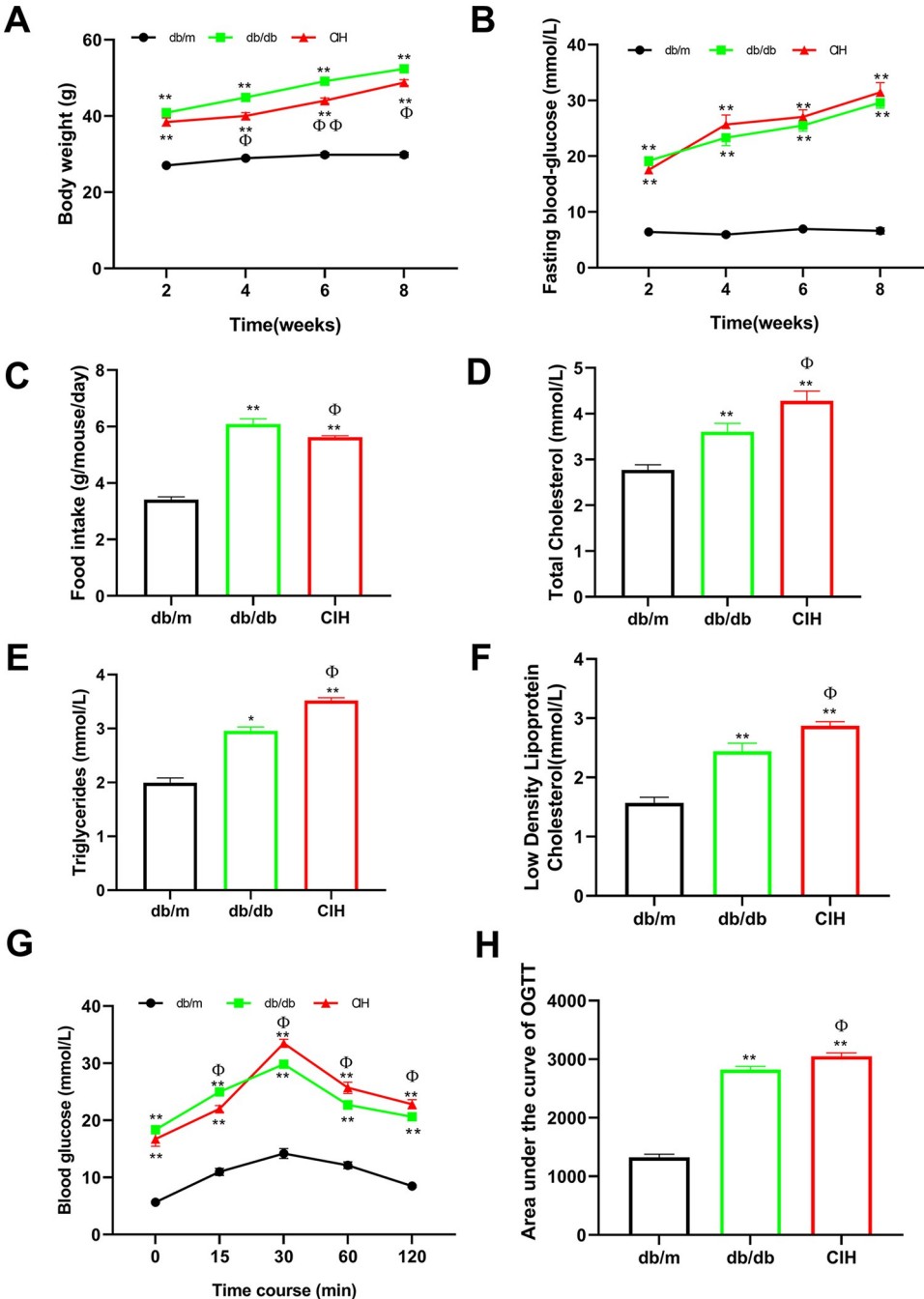

**Fig 1. Effects of CIH exposure on the metabolic characteristics of male db/db mice at 12 to 13 weeks.** (A-C) The levels of BW, FBG and food intake. (D-F) The levels of TC, TG, and LDL-C. (G-H) Glucose tolerance tests and area under curve (AUC) analyses of the OGTT. n = 5. Data are presented as means ± S.E.M. *$p < 0.05$ and **$p < 0.01$ vs. db/m group; $^{\Phi}p < 0.05$ and $^{\Phi\Phi}p < 0.01$ vs. db/db group.

## Effects of CIH exposure on the insulin-signaling pathway of db/db mice

As shown in Fig 2A, compared to the db/m group, the level of HOMA-IR in the db/db group and the CIH group was significantly increased ($p < 0.01$). Compared to the db/db group, the level of HOMA-IR was increased in the CIH group ($p < 0.05$). Compared to the db/m group,

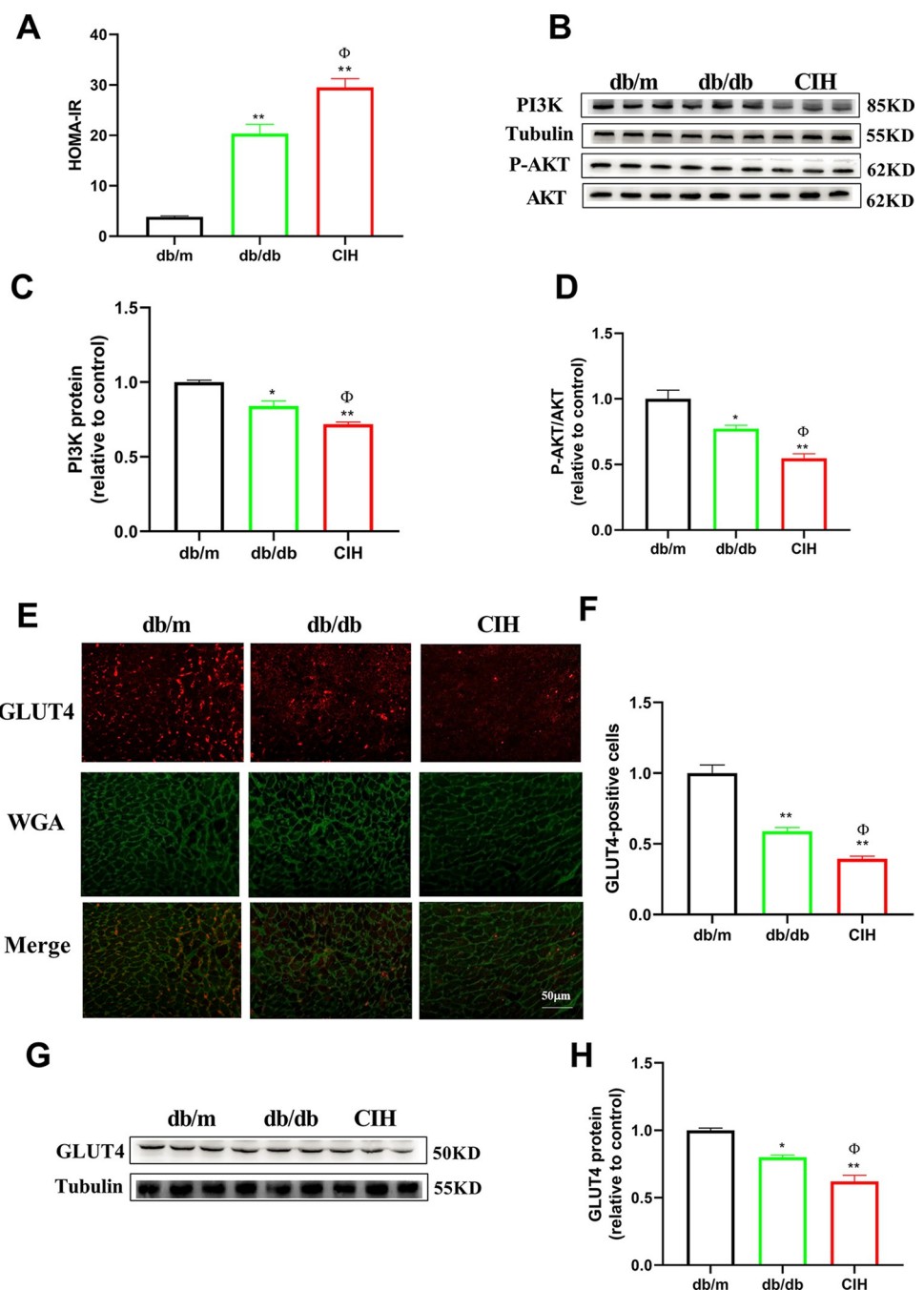

**Fig 2. Effects of CIH exposure on the insulin-signaling pathway of male db/db mice at 12 to 13 weeks.** (A) The level of HOMA-IR. n = 5. (B-D) Representative blot images and quantitative analysis of PI3K and P-AKT. (E-F) Representative images and quantitative analysis of GLUT4 (scale bar, 50 μm). (G-H) Representative blot images and quantitative analysis of GLUT4. n = 3. Data are presented as means ± S.E.M. $*p<0.05$ and $**p < 0.01$ vs. db/m group; $^{\Phi}p < 0.05$ and $^{\Phi\Phi}p < 0.01$ vs. db/db group.

the expression of PI3K and P-AKT was decreased in the db/db group and CIH group ($p < 0.05$, Fig 2B–2D). Compared to the db/db group, the expression of PI3K and P-AKT in the CIH group was decreased ($p < 0.05$). As shown in Fig 2E–2H, the expression of GLUT4 in the CIH group was determined by immunofluorescence and Western blotting. Compared to

the db/m group, the expression of GLUT4 in the db/db group and the CIH group was decreased ($p < 0.05$). Compared to the db/db group, the expression of GLUT4 in the CIH group was decreased ($p < 0.05$).

## Effects of CIH exposure on cardiac pathophysiology of db/db mice

As shown in Fig 3A, compared to the db/m group, the results showed that myocardial disorders in the db/db group and the CIH group. Compared to the db/db group, CIH exposure aggravated the myocardial disorders. Masson staining was used to evaluate the myocardial fibers of db/db mice. Compared to the db/m group, the area of myocardial tissue fibrosis was increased in the db/db group and the CIH group ($p < 0.01$, Fig 3C). Compared to the db/db group, the area of myocardial tissue fibrosis was increased in CIH group ($p < 0.05$). Compared

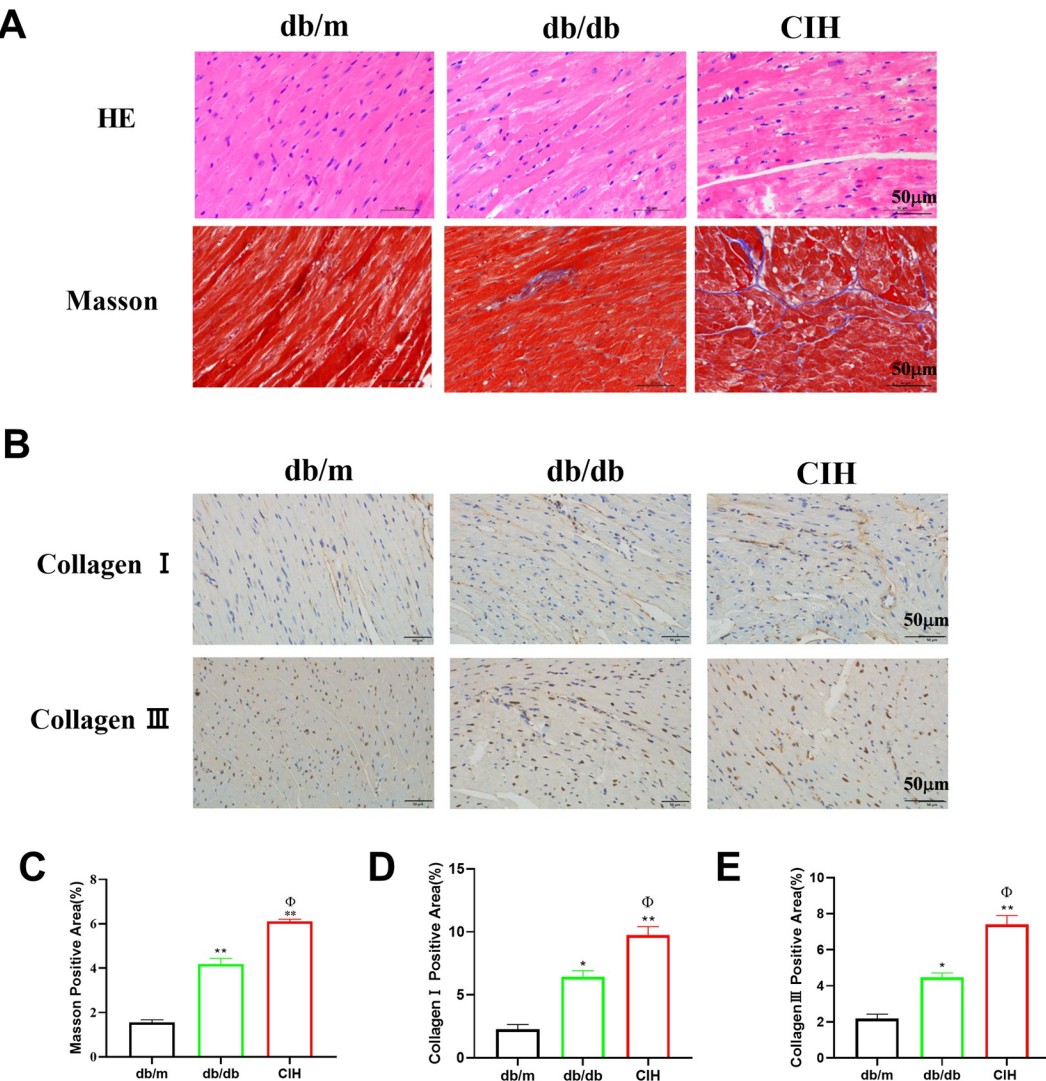

**Fig 3. Effects of CIH exposure on cardiac histopathology of male db/db mice at 12 to 13 weeks.** (A) Representative images of HE staining and Masson staining of cardiac tissues (scale bar, 50 μm). (B) Representative images of collagen I and collagen III protein expression of cardiac tissue (scale bar, 50 μm). (C) The Masson positive area of cardiac tissues. (D-E) The collagen I and collagen III protein positive area of cardiac tissues. n = 3. Data are presented as means ± S.E.M. *$p < 0.05$ and **$p < 0.01$ vs. db/m group; $^{\Phi}p < 0.05$ and $^{\Phi\Phi}p < 0.01$ vs. db/db group.

to the db/m group, the expression of collagen I and collagen III in the db/db group and the CIH group increased ($p < 0.05$, $p < 0.01$, Fig 3B–3E), and that of the CIH group showed a further increase compared to the db/db group ($p < 0.05$).

## Effects of CIH exposure on cardiac function of db/db mice

The effect of CIH exposure on the cardiac function of db/db mice by echocardiography (Fig 4A and 4B). The levels of LVEF and LVFS are important indicators of left ventricular systolic function [20], which is the most commonly used method to evaluate ventricular function. Compared to the db/m group, the LVEF and LVFS of the db/db group and the CIH group were significantly decreased ($p < 0.01$, Fig 4C and 4D). Compared to the db/db group, the LVEF and LVFS of the CIH group were decreased ($p < 0.05$). Compared to the db/m group, the LVDd and LVDs of the db/db group and the CIH group were significantly increased ($p < 0.01$, Fig 4E and 4F). Compared to the db/db group, the LVDd and LVDs of the CIH group were increased ($p < 0.05$). The E/A ratio is an important indicator of left ventricular diastolic function. Compared to the db/m group, the ratio of E/A in the db/db group and the CIH group was significantly decreased ($p < 0.01$, Fig 4G). Compared to the db/db group, the ratio of E/A in the CIH group was decreased ($p < 0.05$). DCM causes an increase in the serum levels of cTnI, LDH, and CK-MB [21]. Consistent with this, compared to the db/m group, the levels of LDH, CK-MB and cTnI in the db/db group and CIH group were increased ($p < 0.05$, Fig 4H–4J). Compared to the db/db group, the levels of LDH, CK-MB and cTnI of the CIH group were increased ($p < 0.05$).

## Effects of CIH exposure on cardiac apoptosis of db/db mice

Compared to the db/m group, the number of positive apoptotic cells in the db/db group and CIH group was significantly increased ($p < 0.01$, Fig 5A and 5B). Compared to the db/db group, the number of positive apoptotic cells in the CIH group was increased ($p < 0.05$). The protein expression of Bax, Bcl-2, and Caspase-3 was detected by western blotting (Fig 5C). Compared to the db/m group, the Bax/Bcl-2 ratio was increased in the db/db group and CIH group ($p < 0.01$, Fig 5D). Compared to the db/db group, the Bax/Bcl-2 ratio in the CIH group was increased ($p < 0.05$). Caspase-3 is a major executive protein of apoptosis and is involved in the apoptosis of diabetic cardiomyocytes [22]. Compared to the db/m group, the cleaved-caspase-3/pro-caspase-3 ratio was significantly increased in the db/db group and the CIH group ($p < 0.05$, Fig 5E). Compared to the db/db group, the cleaved-caspase-3/pro-caspase-3 ratio in the CIH group was increased ($p < 0.05$).

## Effects of CIH exposure on cardiac oxidative stress of db/db mice

As shown in Fig 6A and 6C, compared to the db/m group, the levels of SOD and GSH-Px were decreased in the db/db group and the CIH group ($p < 0.05$). Compared to the db/db group, the levels of SOD and GSH-Px in the CIH group were decreased ($p < 0.05$). Compared to the db/m group, the level of MDA was increased in the db/db group and the CIH group ($p < 0.05$, Fig 6B). Compared to the db/db group, the level of MDA of the CIH group was increased ($p < 0.05$). Moreover, ROS expression in cardiac tissues was detected by DHE (Fig 6D and 6E). Compared to the db/m group, the expression of ROS in the db/db group and CIH group was significantly increased ($p < 0.01$). Compared to the db/db group, the expression of ROS in the CIH group was increased ($p < 0.05$). As shown in Fig 6F, the mitochondrial structure of the db/m group was intact and clear, whereas the mitochondrial structure of the db/db group and CIH group was deformed. Compared to the db/db group, mitochondrial swelling and vacuolar degeneration were observed in the CIH group.

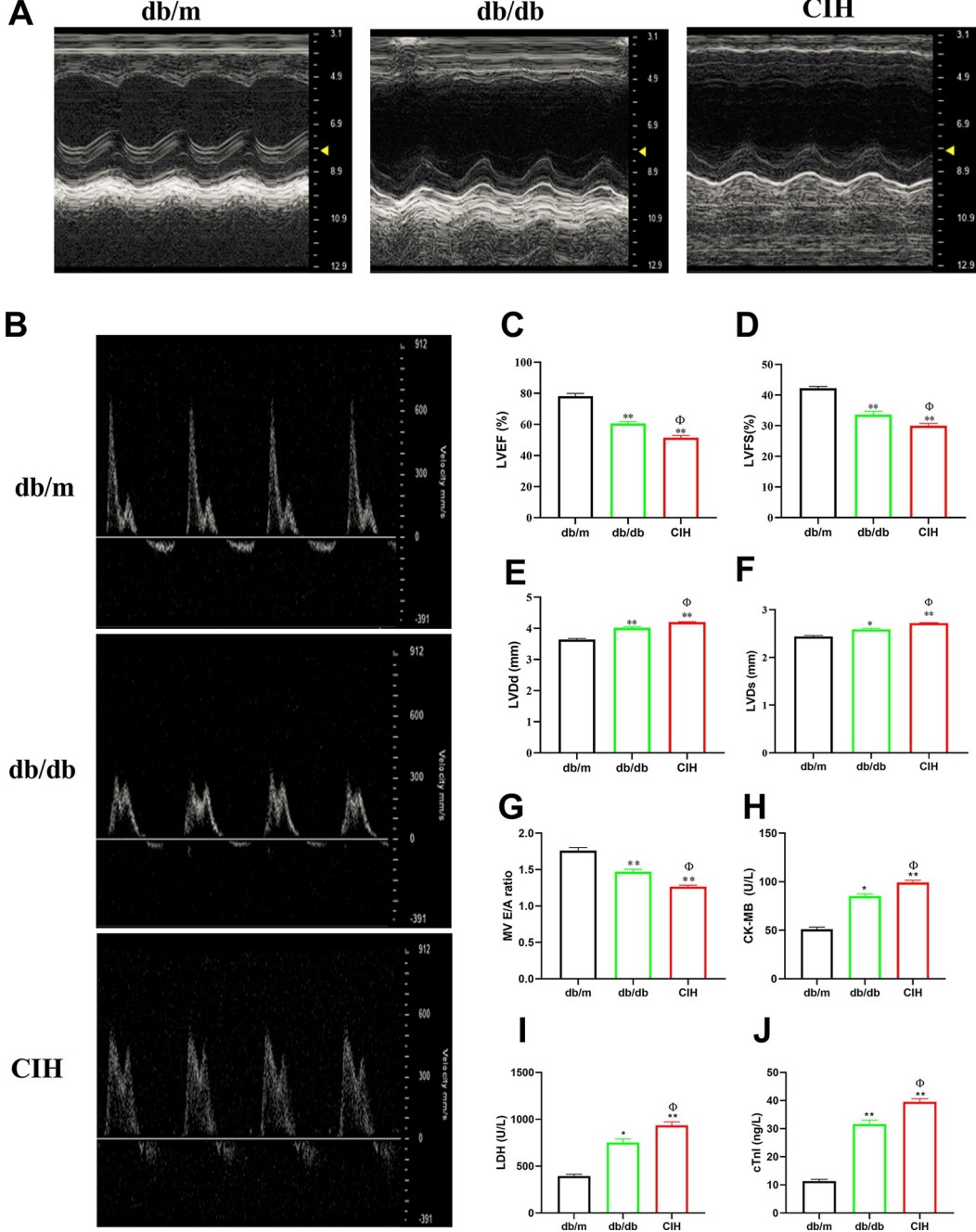

**Fig 4. Effects of CIH exposure on cardiac function of male db/db mice at 12 to 13 weeks.** (A) Representative images of M-mode echocardiography. (B) Representative images of Doppler spectrum. (C–G) The measurement of LVEF, LVFS, LVDd, LVDs and E/A. (H-J) The levels of CK-MB, LDH and cTnI. n = 5. Data are presented as means ± S.E.M. $*p < 0.05$ and $**p < 0.01$ vs. db/m group; $^{\Phi}p < 0.05$ and $^{\Phi\Phi}p < 0.01$ vs. db/db group.

### Effects of CIH exposure on the LKB1/AMPK/Nrf2 signaling pathway of db/db mice

The AMPK/Nrf2/HO-1 pathway has been implicated in oxidative stress damage in the heart and LKB1 is an upstream kinase of AMPK. Studies have confirmed that oxidative stress

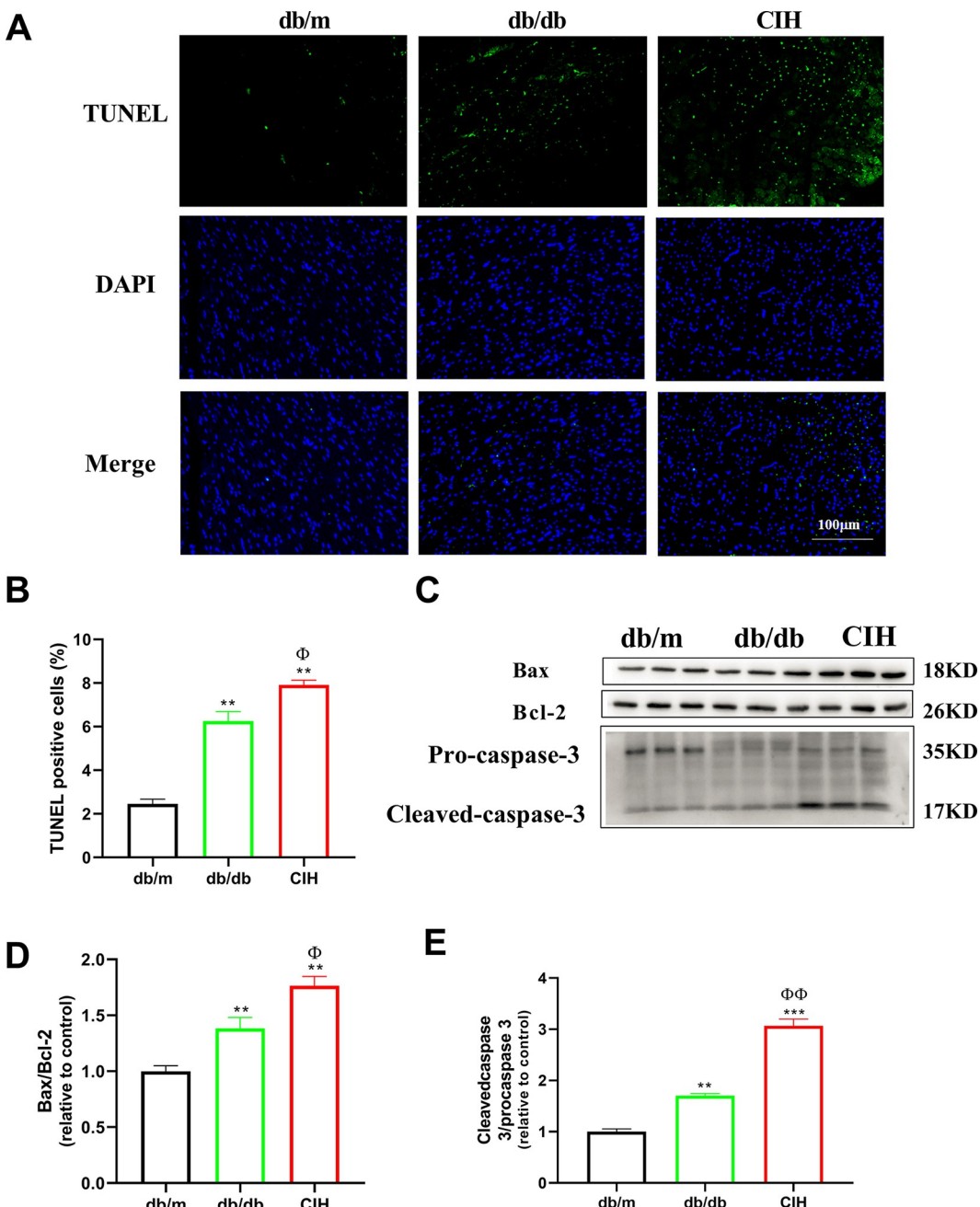

**Fig 5. Effects of CIH exposure on cardiac apoptosis of male db/db mice at 12 to 13 weeks.** (A) Representative images of TUNEL staining (scale bar, 100 μm). (B)The measurement of cardiac apoptosis. (C) Representative blot images of Bax, Bcl-2, and Caspase-3. (D-E) The protein expression and quantitative analysis of Bax, Bcl-2, Caspase-3. n = 3. Data are presented as means ± S.E.M. $^{*}p < 0.05$, $^{**}p < 0.01$, $^{***}P < 0.001$ vs. db/m group; $^{\Phi}p < 0.05$ and $^{\Phi\Phi}p < 0.01$ vs. db/db group.

reduces LKB1 activity, which in turn reduces AMPK activity [23]. Compared to the db/m group, the p-LKB1/LKB1 and p-AMPK/AMPK ratio in the db/db group and CIH group was decreased ($p < 0.05$, Fig 7A–7C). Compared to the db/db group, the ratio of p-LKB1/LKB1 and p-AMPK/AMPK in the CIH group was decreased ($p < 0.05$). Compared to the db/m group, the expression of Nrf2 and HO-1 in the db/db group and the CIH group was decreased ($p < 0.05$, Fig 7D and 7E). Compared to the db/db group, the protein expression of Nrf2 and

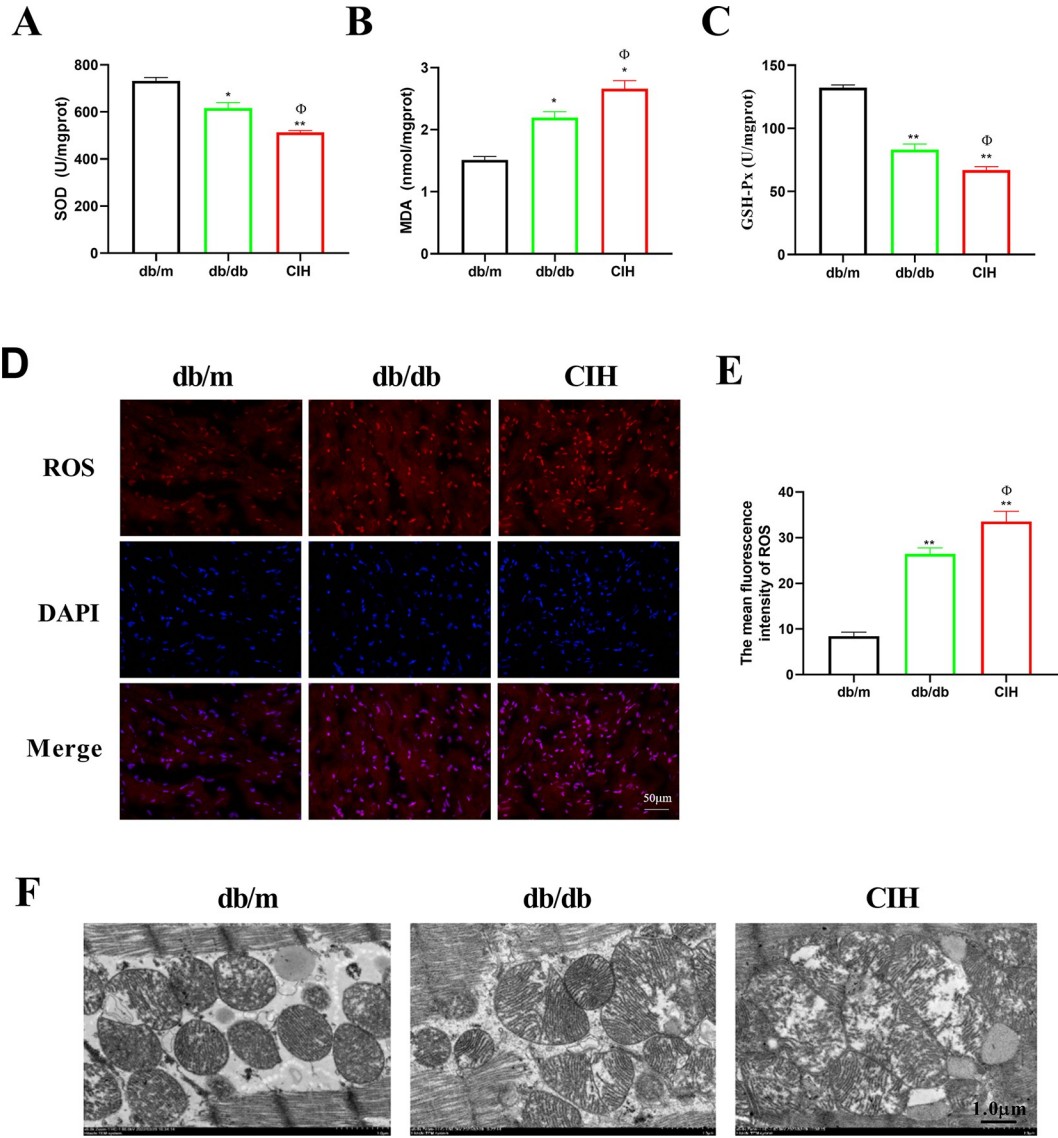

**Fig 6. Effects of CIH exposure on cardiac oxidative stress of male db/db mice at 12 to 13 weeks.** (A-C) The levels of SOD, MDA and GSH-Px. n = 5. (D-E) Representative images of ROS and the expression of ROS (scale bar, 50 μm). (F) Representative images of electron micrographs (scale bar, 1.0 μm). n = 3. Data are presented as means ± S.E.M. $*p < 0.05$ and $**p < 0.01$ vs. db/m group; $^{\Phi}p < 0.05$ and $^{\Phi\Phi}p < 0.01$ vs. db/db group.

HO-1 in the CIH group was decreased ($p < 0.05$). Furthermore, the expression of nuclear-Nrf2 was detected by immunofluorescence and western blot (Fig 7F–7I). Compared to the db/m group, the expression of nuclear-Nrf2 in the db/db group and the CIH group was decreased ($p < 0.01$). Compared to the db/db group, the expression of nuclear-Nrf2 in the CIH group was decreased ($p < 0.05$).

## Effects of IH and HG on the cell viability, ROS production, and mitochondrial membrane potential of H9C2 cells

As shown in Fig 8A and 8B, cells were used treated with different concentrations of HG for 24 and 48 h; we selected a final concentration of 30mM HG for 48h. The IH+HG group was

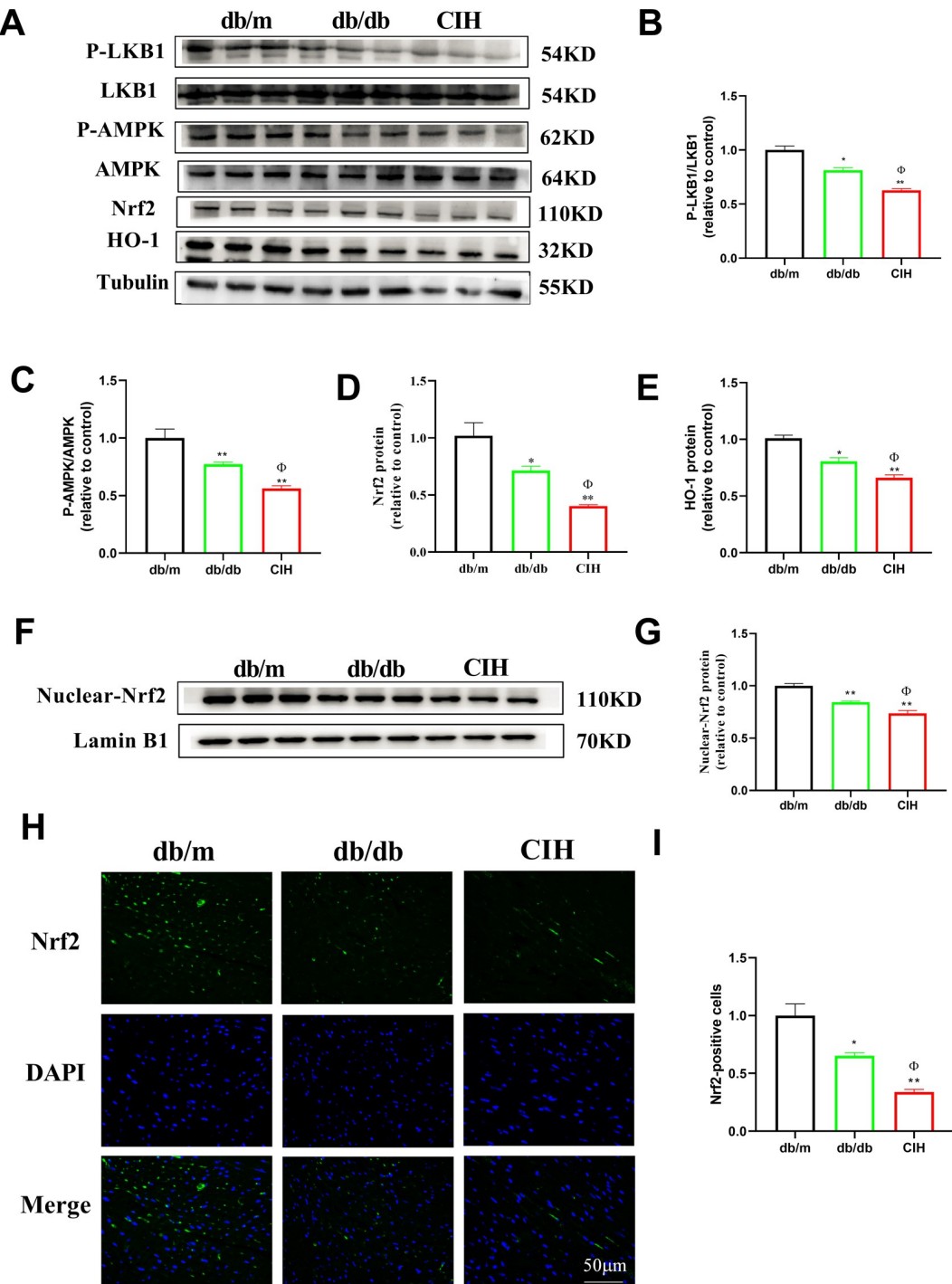

**Fig 7. Effects of CIH exposure on the LKB1/AMPK/Nrf2 signaling pathway of male db/db mice at 12 to 13 weeks.** (A-E) Representative blot images and quantitative analysis of LKB1, AMPK, Nrf2, HO-1. (F-G) Representative blot images and quantitative analysis of Nuclear-Nrf2. (H-I) Representative images of Nuclear-Nrf2 and the quantitative analysis for Nuclear-Nrf2 (scale bar, 100 μm). n = 3. Data are presented as means ± S.E.M. *$p < 0.05$ and **$p < 0.01$ vs. db/m group; $^{\Phi}p < 0.05$ and $^{\Phi\Phi}p < 0.01$ vs. db/db group.

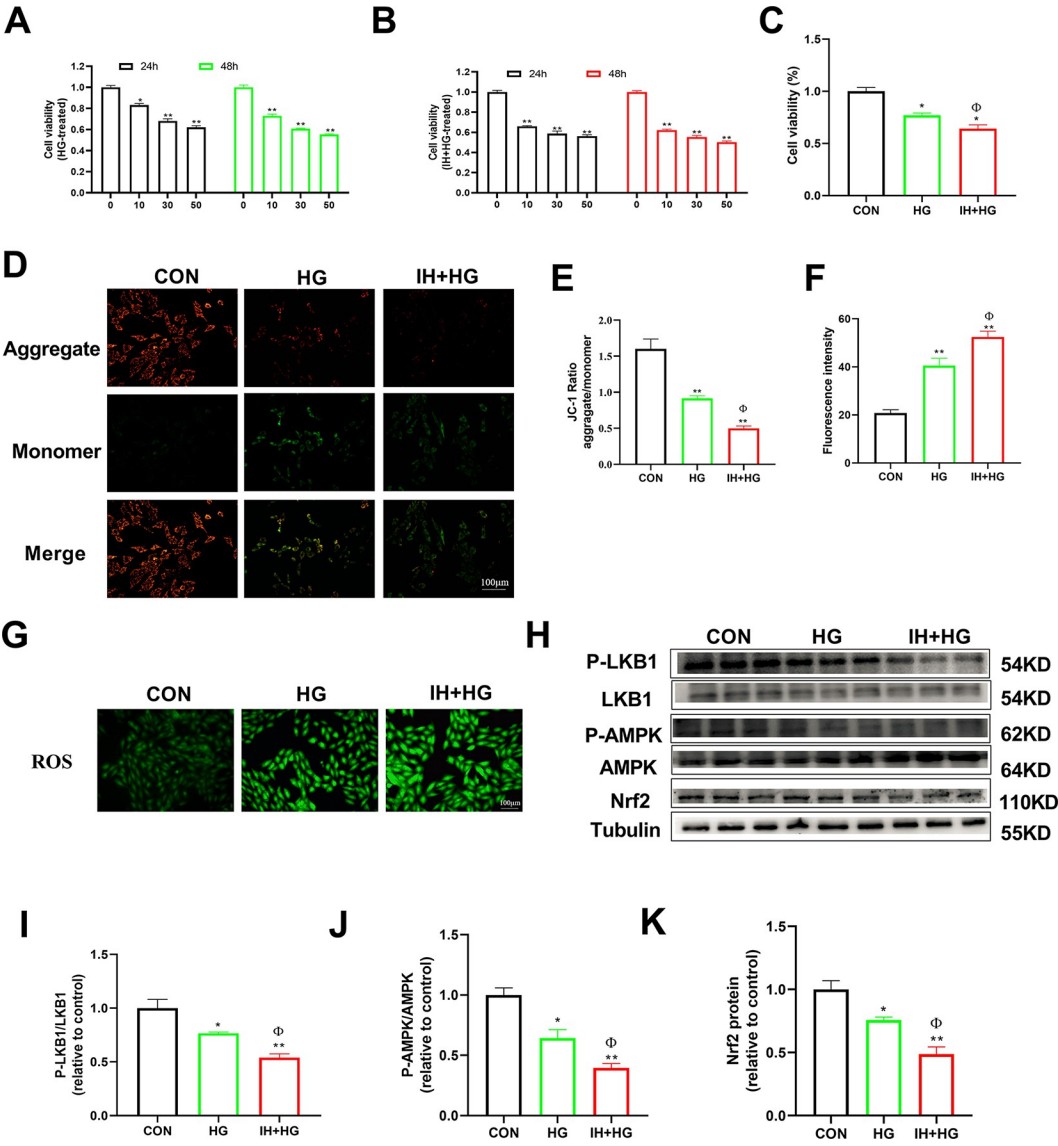

**Fig 8. Effects of IH and HG on the cell viability, ROS production, and mitochondrial membrane potential of H9C2 cells.**
(A) The cell viability of H9C2 cells was treated with HG with 0, 10, 30, and 50mM for 24 and 48 h. (B) The H9C2 cells were treated with IH and HG for 24 and 48 h, and cell viability of H9C2 cells was measured. (C) The cell viability of H9C2 cells treated with IH and HG. n = 6. (D-E) Representative images of mitochondrial membrane potential and quantitative analysis of the JC-1 ratio. n = 4. (F-G) Representative images of ROS and quantitative analysis of ROS. (H-K) Representative blot images and quantitative analysis of LKB1, AMPK, and Nrf2. n = 3. Data are presented as means ± S.E.M. $^*p < 0.05$ and $^{**}p < 0.01$ vs. CON group; $^{\Phi}p < 0.05$ and $^{\Phi\Phi}p < 0.01$ vs. HG group.

treated with IH for 48h, compared to the CON group, and the cell viability of the HG group and the IH+HG group was decreased ($p < 0.05$, Fig 8C). Compared to the HG group, the cell viability of the IH+HG group was decreased ($p < 0.05$). As shown in Fig 8D and 8E, compared to the CON group, the cell membrane potential of the HG group and the IH+HG group was significantly decreased ($p < 0.01$). Compared to the HG group, the cell membrane potential of the IH+HG group was decreased ($p < 0.05$). As shown in Fig 8F and 8G, compared to the CON group, the expression of ROS in the HG group and the IH+HG group was significantly increased ($p < 0.01$). Compared to the HG group, the expression of ROS in the IH+HG group

was increased ($p < 0.05$). As shown in Fig 8H–8K, compared to the CON group, the expression of p-LKB1/LKB1 and p-AMPK/AMPK and Nrf2 in the HG group and IH+HG group was decreased ($p < 0.05$, $p < 0.01$). Compared to the HG group, the expression of p-LKB1/LKB1 and p-AMPK/AMPK and Nrf2 in IH+HG group was decreased ($p < 0.05$).

### Effects of IH and HG on the LKB1/AMPK/Nrf2 signaling pathway of H9C2 cells

As shown in Fig 9A and 9B, compared to the CON group, the cell membrane potential of the IH +HG group was decreased ($p < 0.01$). Compared to the IH+HG group, the cell membrane potential of the Metformin group and the A-769662 group was increased ($p < 0.01$). As shown in Fig 9C and 9D, compared to the CON group, the expression of ROS in the IH+HG group was significantly increased ($p < 0.01$). Compared to the IH+HG group, the expression of ROS in the Metformin group and the A-769662 group was decreased ($p < 0.05$). As shown in Fig 9E–9G, compared to the CON group, the expression of p-AMPK/AMPK and Nrf2 in the IH+HG group was decreased ($p < 0.05$). Compared to the IH+HG group, the expression of p-AMPK/AMPK and Nrf2 in the Metformin group and the A-769662 group was increased ($p < 0.05$).

## Discussion

This study mainly discussed the mechanism by which CIH aggravates IR and glycolipid metabolism disorders through oxidative stress, inducing myocardial cell apoptosis, leading to cardiac dysfunction, and further resulting in the development and deterioration of DCM. It provides a new method for the pathogenesis of OSA with DCM.

The db/db mice are congenital obese type 2 DCM mice with leptin deficiency caused by gene mutation [24], characterized by hyperinsulinemia, hyperlipidemia and myocardial hypertrophy [25], which has been widely used to study the metabolic pathogenesis and activated inflammatory mechanisms of type 2 DCM [26]. As a common respiratory disease, OSA often coexists with T2DM, and can promote the disease progression of T2DM and its complications [27]. CIH is the core pathological basis of OSA, associated with IR and glycolipid metabolism disorders in OSA patients, which is an important risk factor for cardiovascular diseases. Clinical studies suggest that OSA can increase the level of blood glucose, decreasing insulin sensitivity and aggravating IR in T2DM [28, 29]. In our experiment, after 8 weeks of CIH exposure, db/db mice had an increase in the fluctuation of blood glucose and impaired glucose tolerance, which is a potential risk factor for accelerating the occurrence and development of DCM.

Myocardial disorders and myocardial fibrosis are the characteristic pathological changes of DCM myocardial remodeling. Recently, it was shown that the hypoxia-induced inflammatory response aggravates myocardial cell damage [30]. Under the stimulation of inflammatory factors, abnormal deposition of extracellular matrix and increased content of collagen fibers jointly promote the occurrence and development of myocardial fibrosis, and ultimately participate in the destruction of cardiac structure and function [31]. Our results showed that collagen fibers were increased in CIH group, suggesting that CIH exposure may aggravate the progression of myocardial fibrosis in DCM. To assess the association between CIH and cardiac dysfunction of DCM, echocardiography was used to measure cardiac function. Goes CM found that repeated episodes of hypoxia can lead to left ventricular dysfunction [32]. Our results showed that CIH exposure could cause left ventricular systolic and diastolic dysfunction in db/db mice. The level of myocardial enzyme is used to measure the damage degree of myocardial cells indirectly. We found that the serum levels of CK-MB, LDH and cTnI were significantly reduced after CIH exposure, suggesting that CIH further aggravated myocardial injury in DCM.

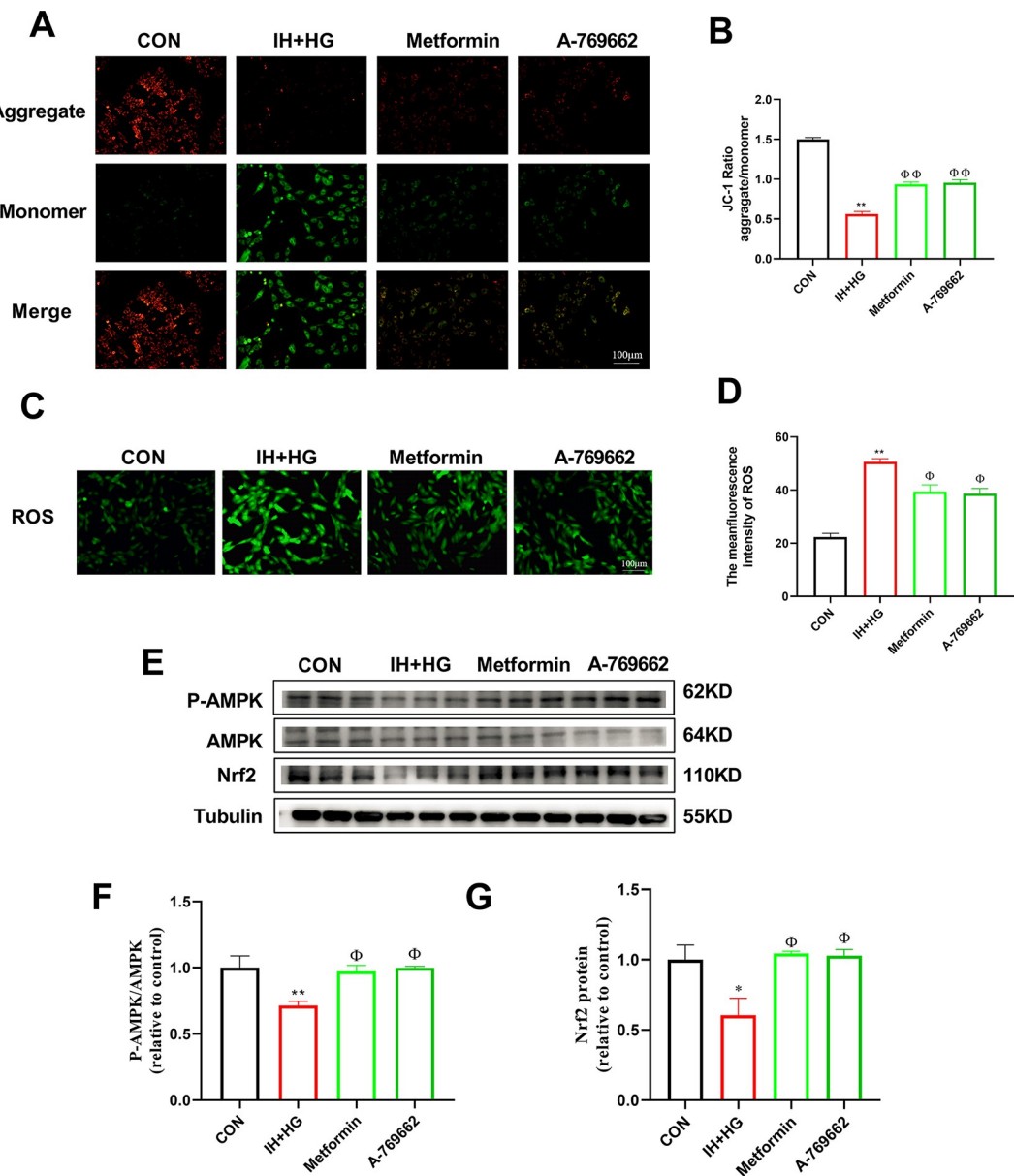

**Fig 9. Effects of IH and HG on the LKB1/AMPK/Nrf2 signaling pathway of H9C2 cells.** (A-B) Representative images of mitochondrial membrane potential and quantitative analysis of the JC-1 ratio. (C-D) Representative images and quantitative analysis of ROS. (E-G) Representative blot images and quantitative analysis of AMPK and Nrf2. n = 3. Data are presented as means ± S.E.M. $*p < 0.05$ and $**p < 0.01$ vs. CON group; $^{\Phi}p < 0.05$ and $^{\Phi\Phi}p < 0.01$ vs. IH+HG group.

Oxidative stress injury of myocardial tissue is a key factor in the occurrence and development of DCM [33]. Oxidative stress injury is the main pathological mechanism leading to the occurrence and progression of DCM [34]. CIH is associated with increased ROS and oxidative stress [35]. Oxidative stress is up-regulated in OSA patients and may adversely affect cardiovascular disease [36]. ROS in the human heart is derived from mitochondrial production and NADPH [37]. ROS can cause molecular damage, leading to mitochondrial dysfunction and reducing the oxidative capacity of fatty acids, which results in lipid accumulation, fibrosis, diastolic dysfunction, and even heart failure, all of which are related to the occurrence of DCM [38, 39]. The mitochondrial structure of db/db mice is swollen and irregular, suggesting CIH

exposure aggravating the mitochondrial structural damage, increasing the production of ROS and promoting the progression of DCM. Previous studies have shown that under IH conditions, H9C2 cardiomyocytes are also damaged, ROS is markedly increased, oxidative stress in cardiomyocytes is increased, mitochondria are seriously fragmented, the membrane potential is decreased, and the mitochondrial function is impaired. Together, these data show that CIH has serious damaging effects on db/db mice and H9C2 cardiomyocytes [40–42].

Glucose metabolism disorders are correlated with oxidative stress, which reduces antioxidant levels and further aggravates oxidative stress. Oxidative stress is related to the activation of PI3K/AKT/GLUT4 signaling pathway. Under the condition of IR, the utilization of glucose by the heart is significantly reduced, which makes the heart more dependent on fatty acid oxidation for energy metabolism. GLUT4 is mainly expressed in skeletal muscle, myocardium and fat cells. The stimulation of GLUT4 transportation will be an important step in improving cardiac glucose metabolism. CIH could increase the level of oxidative stress and block the activation of PI3K/AKT/GLUT4 signaling pathway, further aggravating glucose metabolism disorders and oxidative stress damage of myocardial cells, and ultimately affecting the cardiac function of DCM [43]. Studies have shown that CIH exposure could reduce the mRNA and protein levels of GLUT4 in skeletal muscle, which may occur simultaneously in liver, adipose tissue and even all tissues of the body [44]. After CIH exposure, our results showed that the expression of GLUT4 was significantly reduced, aggravating glucose metabolism disorders, oxidative stress, and suggesting an increase in the cardiac dysfunction of DCM.

Myocardial apoptosis is an important mechanism of myocardial dysfunction in DCM [45]. Myocardial apoptosis is the main cause of myocardial cell loss, myocardial remodeling and dysfunction in diabetic animal models and patients [22]. Caspase-3 is an important cysteine protease that participates in apoptosis directly after activation. We found that TUNEL positive cells, the ratio of cleaved-caspase-3/pro-caspase-3 and Bax/Bcl-2 were significantly increased after exposure to CIH. Consistent with previous studies [46–48], in this study, IH-induced H9C2 cardiomyocyte injury was closely related to decreasing the expression of anti-apoptotic protein and increasing the expression of pro-apoptotic protein, suggesting that CIH exposure can increase myocardial apoptosis in DCM.

Nrf2 is one of the therapeutic targets for DCM, which plays an important role in enhancing myocardial antioxidant, anti-inflammatory, anti-fibrosis and anti-apoptosis abilities. Several studies have shown that Nrf2 can be activated by AMPK. The activation of Nrf2 can activate downstream HO-1, which quickly and effectively removed excessive ROS, reducing the myocardial injury of DCM. AMPK is a central protein that coordinates metabolism and energy and regulates a variety of lipid metabolization-related enzymes and transcription factors. The AMPK/Nrf2 signaling pathway plays an important role in reducing oxidative stress response and inhibiting mitochondria mediated apoptosis [49]. LKB1, as the upstream kinases of AMPK, has been proven to activate AMPK. The expression of p-LKB1 is positively correlated with AMPK activity. We found that CIH exposure reduced the expression of p-LKB1, p-AMPK and Nrf2 in vitro and in vivo, suggesting that CIH exposure may aggravate oxidative stress and promote glycolipid metabolism disorders by inhibiting LKB1/AMPK/Nrf2 signaling pathway, and further aggravate the cardiac function impairment of DCM. Therefore, the inhibition of oxidative stress and apoptosis as therapeutic targets is crucial for the prevention and treatment of DCM.

## Conclusion

In conclusion, CIH could aggravate oxidative stress, promoting glycolipid metabolism disorders and leading to myocardial apoptosis by inhibiting cardiac LKB1/AMPK/Nrf2 signaling pathway in vitro and in vivo, further aggravating cardiac function injury of DCM (Fig 10).

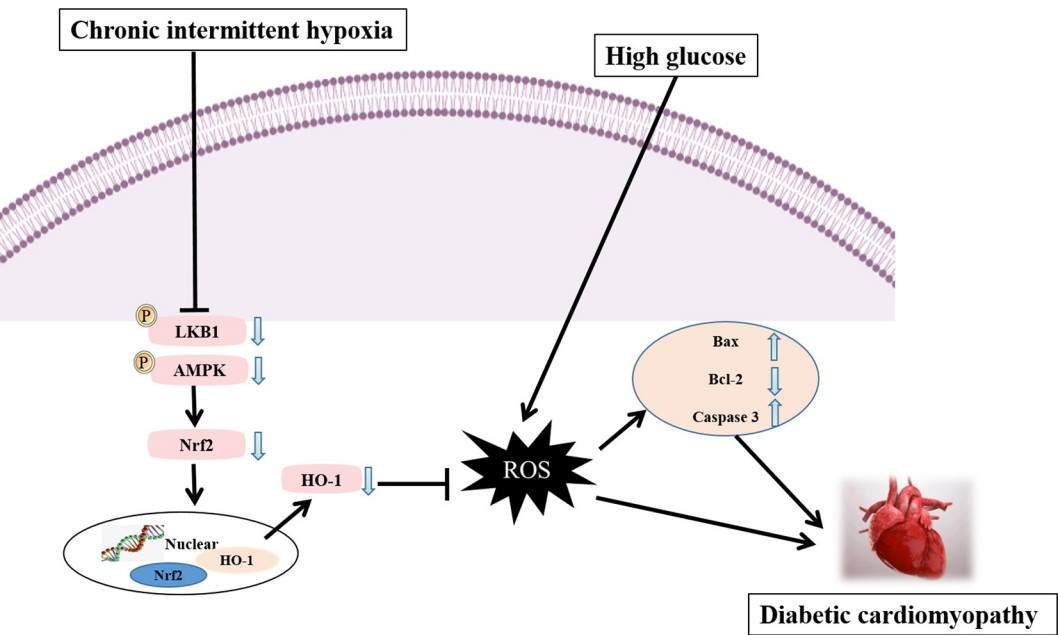

**Fig 10. The possible mechanism diagram of DCM is aggravated by CIH.**

## Supporting information

**S1 Graphical abstract.**
(PDF)

**S1 Raw images.**
(PDF)

**S1 Data.**
(XLSX)

## Author Contributions

**Conceptualization:** Ensheng Ji, Shengchang Yang.

**Data curation:** Bingbing Liu, Jianchao Si.

**Formal analysis:** Dongli Li, Tingting Li, Yi Tang.

**Funding acquisition:** Ensheng Ji, Shengchang Yang.

**Investigation:** Dongli Li, Tingting Li, Yi Tang.

**Methodology:** Bingbing Liu.

**Software:** Kerong Qi.

**Visualization:** Jianchao Si.

**Writing – original draft:** Bingbing Liu, Jianchao Si, Kerong Qi.

**Writing – review & editing:** Ensheng Ji, Shengchang Yang.

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
