## [Decision Letter · Decision Letter 0]

20 Oct 2023

PONE-D-23-28549Chronic intermittent hypoxia aggravated diabetic cardiomyopathy through LKB1/AMPK/Nrf2 signaling pathwayPLOS ONE

Dear Dr. Yang,

Thank you for submitting your manuscript to PLOS ONE. After careful consideration, we feel that it has merit but does not fully meet PLOS ONE’s publication criteria as it currently stands. Therefore, we invite you to submit a revised version of the manuscript that addresses the points raised during the review process.

We look forward to receiving your revised manuscript.

Kind regards,

Michael Bader

Academic Editor

PLOS ONE

Journal Requirements:

"This work was supported by the National Natural Science Foundation of China (82274617), the Hebei Natural Science Foundation (H2022423352, H2022423370), the Science and Technology Project of Hebei Education Department (QN2023159) and the Medical Science Research Projects of Health Commission of Hebei Province (20231573)."

Reviewers' comments:

Reviewer's Responses to Questions

**Comments to the Author**

1. Is the manuscript technically sound, and do the data support the conclusions?

Reviewer #1: Yes

Reviewer #2: Partly

2. Has the statistical analysis been performed appropriately and rigorously? 

Reviewer #1: I Don't Know

Reviewer #2: No

3. Have the authors made all data underlying the findings in their manuscript fully available?

Reviewer #1: Yes

Reviewer #2: Yes

4. Is the manuscript presented in an intelligible fashion and written in standard English?

Reviewer #1: Yes

Reviewer #2: No

5. Review Comments to the Author

Reviewer #1: Although chronic intermittent hypoxia (CIH) exposure promotes glycolipid metabolism disorders and myocardial apoptosis, aggravating the myocardial injury via liver kinase B1 (LKB1)/5' adenosine monophosphate-activated protein kinas (AMPK)/nuclear erythroid 2-related factor 2 (Nrf2) pathway in vitro and in vivo is interesting, numbers of points need clarifying and certain statements require further justification. These are given below.

<point>

1. The authors described, “All procedures involving animal and experimental protocols were approved by the Hebei University of Chinese Medicine Animal Care and Use Committee.” (lines 99–102) without showing approval date and number(s).

2. In Fig. 1, what age and gender of mice were used should be clarified.

3. In Fig. 2, what age and gender of mice were used should be clarified.

4. In Fig. 3, what age and gender of mice were used should be clarified.

5. In Fig. 4, what age and gender of mice were used should be clarified.

6. In Fig. 5, what age and gender of mice were used should be clarified.

7. In Fig. 6, what age and gender of mice were used should be clarified. In addition, scale bar(s) should be added in Fig. 6F.

8. In Fig. 7, what age and gender of mice were used should be clarified.

9. In Fig. 8, what age and gender of mice were used should be clarified.

10. In Suppl. Fig. 2, the arrow did not indicate any band/signal in Tubulin “55KD”.

11. In Suppl. Fig. 7, “dh/m”, “db/db”, and “CIH” did not match the band position. The 110 KD band in “Nuclear-Nrf2” did not fit in the figure.

12. In Suppl. Fig. 8, the 54 KD band(s) in “p-LKB1” were not clear.

13. “Dihydroethidium” (line 35) should be changed to “dihydroethidium”.

14. “Ismail Laher [9]” (line 66) should be changed to “Baran et al. [9]” or “Laher and his colleagues [9]”.

15. “Hu k [10]” (line 69) should be changed to “Peng and Hu [10]”.

16. “37°C” (lines 153, 160, and 165) should be changed to “37 ˚C”.

17. Concerning intermittent hypoxia in cardiomyocytes, several paperers (Hu, Z. et al. Sleep Breath doi: 10.1007/s11325-023-02863-8. Online ahead of print; Chen, Q. et al. Sleep Breath 27, 129-136, 2023; Dong, N. & Liu, W.Y. Sleep Breath 27, 399-410; Takasawa, S. et al. Int. J. Mol. Sci. 23, 8782, 2022; Takasawa, S. et al. Int. J. Mol. Sci. 23, 12414, 2022; Moulin, S. et al. Antioxidants 11, 1462, 2022) have recently been reported. The authors should cite the papers and make some discussions.

18. There are some reports concerning human cardiomyocytes in intermittent hypoxia such as Regev, D. et al. Int. J. Mol. Sci. 23, 10272, 2022, Huang, J. et al. Sleep Breath 27, 1005-1011, 2023; Moulin, S. et al. Antioxidants 11, 1462, 2022; Chen, Q. et al. Sleep Breath 27, 129-136, 2023; Dong, N. & Liu, W.Y. Sleep Breath 27, 399-410). The authors should cite the papers and make some discussions.</point>

Reviewer #2: Liu et al. reported on the effects of chronic intermittent hypoxia (CIH) on diabetic cardiomyopathy. Using in vivo and in vitro models they concluded that CIH aggravated diabetic cardiomyopathy through LKB1/AMPK/Nrf2 signaling pathway resulting in cardiomyocyte apoptosis and increased myocardial fibrosis. Overall, it is a relevant topic, as sleep perturbations and diabetes are highly prevalent clinical disorders which may negatively influence each other and understanding of pathomechanisms underlying these interactions is of great importance. However, this paper needs to be improved, as it lacks some important information, especially in the method section and the presentation of the results is sometimes unclear.

Below please find more detailed comments:

- Introduction: Page 11: “…and excessive ROS can inhibit AMPK inactivation by inhibiting the phosphorylation of LKB1”. Here, I think you meant ROS can inhibit AMPK activation, or?

- Methods, page 11: only male animals were used. Was there any reason not to include females? Including only males is a limitation of this study. How old were the mice at the beginning of the experiment?

- Methods, page 11: “24 db/db mice were randomly divided into three groups (n=8): db/m group, db/db group, and CIH group”. Db/db mice cannot serve as a source of db/m mice. Please rewrite this sentence.

- Methods, page 11: “The concentration of oxygen in the chamber for the CIH group was changed from 21% to 5%, 8 h/day for 8 weeks”. -> How long was one cycle of normoxia-hypoxia-normoxia? Did you measure the extent of hypoxemia in blood of mice under intermittent hypoxia?

- Methods, page 12: How did the authors measure glucose levels?

- Methods, page 13: the authors describe paraffin embedment of heart specimens for subsequent histological stainings, but in the next method section, frozen sections are mentioned. The method of frozen section preparation is missing.

- Methods, page 12: echocardiographic measurements are not precisely described: what view was used? What mode was used? E.g. M-mode for chamber diameters, B-mode for Doppler measurements.

- Methods, page 12: Did you measure thickness of the left ventricular walls and the potential impact of diabetes and IH on this parameter?

- Methods: The authors present results for CK-MB, troponin and LDH serum levels, the corresponding section in the Methods how these parameters were measured is missing.

- Methods, page 15: “we cultured the H9C2 cells with different concentrations of HG for 48 h.” -> what were the concentrations of HG?

- Methods, page 17: Statistical analysis: did the authors check for normal distribution? What method was used for the comparisons between two groups?

- Results, page 23: the results showed that metformin improved membrane potential and decreased oxidative stress in cells which was associated with increased p-AMPK/AMPK and Nrf2 levels. Did this have any impact on the cell viability/apoptosis?

- Graphical abstract: according to the findings, not only cardiomyocyte apoptosis, but also aggravated fibrosis may lead to cardiac dysfunction; fibrosis is not illustrated here.

- Throughout the whole manuscript there are sentences which can hardly be understood (below only a couple of examples of them). Please improve this as it might be very confusing for the reader.

Results, page 17: “Compared to the db/db group, the BW of the CIH group was decreased (p<0.05), but there was no statistical significance in FBG and the fluctuation of blood glucose in the CIH group was increased.” -> This sentence is unclear.

Results, page 19: “As shown in Fig.3A, the results showed that myocardial disorders in the db/db group and the CIH group compared to the db/m group.” -> This sentence is unclear.

Figure 8, legend: “The cell viability of H9C2 cells was treated with IH and HG for 24 h and 48h.” -> This sentence is unclear.

6. PLOS authors have the option to publish the peer review history of their article (what does this mean?). If published, this will include your full peer review and any attached files.

Reviewer #1: No

Reviewer #2: No

---

## [Author Response · Author response to Decision Letter 0]

30 Nov 2023

Response

Dear Professor:

Thank you very much for your comments and professional advice. These opinions help to improve the academic rigor of our article. Based on your suggestion and request, we have made explanations. Thank you very much for providing a new perspective for our research. We appreciate your valuable comments and hope to receive your approval, which we wish to be considered for publication as an original article in “PLOS ONE”.

Journal Requirements:

Question 1: Please ensure that your manuscript meets PLOS ONE's style requirements, including those for file naming. 

Answer: Thanks for your kind suggestions. According to the style requirements of PLOS ONE, we have made revisions to the manuscript. The detailed modifications are as follows:

1. The level 1 heading for all major sections (Abstract, Introduction, Materials and methods, Results, Discussion, Conclusions, References) have been revised to 18pt font and bold type marked in red.

2. The level 2 headings for sub-sections of major sections have been revised to 16pt font and bold type marked in red.

3. This manuscript has been revised to double-space paragraph format.

4. The figure files have been revised to “Fig1 .tif”, “Fig2 .tif”, “Fig3.tif”, “Fig4 .tif”, “Fig5 .tif”, “Fig6 .tif”, “Fig7 .tif”, “Fig8 .tif”, “Fig9 .tif”, “Fig10 .tif”.

5. In aspect of “Figure Citations”, we made a detailed revision in the “Results” of the manuscript and marked in red.

6. In aspect of “References”, we made a detailed revision in the “References” of the manuscript and marked in red. 

7. We added the part of “Supporting information” in the manuscript and marked in red.

Question 2: Thank you for stating the following financial disclosure: 

"This work was supported by the National Natural Science Foundation of China (82274617), the Hebei Natural Science Foundation (H2022423352, H2022423370), the Science and Technology Project of Hebei Education Department (QN2023159) and the Medical Science Research Projects of Health Commission of Hebei Province (20231573)."

Please state what role the funders took in the study. If the funders had no role, please state: ""The funders had no role in study design, data collection and analysis, decision to publish, or preparation of the manuscript."" If this statement is not correct you must amend it as needed.

Answer: Thanks for your kind suggestions. Thank you for reminding us that we have added the roles of funders Shengchang Yang and Ensheng Ji marked in red. Funders Shengchang Yang and Ensheng Ji were involved in the data curation, formal analysis and project administration in the study. Thank you very much for your valuable advice.

Answer: Thanks for your kind suggestions. Thank you for reminding us that we have added the role of funder statement in cover letter marked in red. Thank you very much for your valuable advice.

Question 3: PLOS requires an ORCID iD for the corresponding author in Editorial Manager on papers submitted after December 6th, 2016. Please ensure that you have an ORCID iD and that it is validated in Editorial Manager. To do this, go to ‘Update my Information’ (in the upper left-hand corner of the main menu), and click on the Fetch/Validate link next to the ORCID field. This will take you to the ORCID site and allow you to create a new iD or authenticate a pre-existing iD in Editorial Manager. Please see the following video for instructions on linking an ORCID iD to your Editorial Manager account: https://www.youtube.com/watch?v=_xcclfuvtxQ？？

Answer: Thanks for your kind suggestions. We set up an ORCID iD, thank you.

ORCID iD for the corresponding authors: 

Shengchang Yang yscdekaoyan@163.com 0000-0002-2002-4945

Ensheng Ji jesphy@126.com 0000-0002-5858-7567 

Question 4: Please include your full ethics statement in the ‘Methods’ section of your manuscript file. In your statement, please include the full name of the IRB or ethics committee who approved or waived your study, as well as whether or not you obtained informed written or verbal consent. If consent was waived for your study, please include this information in your statement as well.

Answer: Thanks for your kind suggestions. Thank you for reminding us that we have added the full name of ethics committee and ethical license number in “Methods” marked in red. All procedures involving animal and experimental protocols were approved by the Animal Care and Use Committee of Medical Ethics of Hebei University of Chinese Medicine, and obtained informed written. Thank you very much for your valuable advice.

Question 5: PLOS ONE now requires that authors provide the original uncropped and unadjusted images underlying all blot or gel results reported in a submission’s figures or Supporting Information files. This policy and the journal’s other requirements for blot/gel reporting and figure preparation are described in detail at https://journals.plos.org/plosone/s/figures#loc-blot-and-gel-reporting-requirements and https://journals.plos.org/plosone/s/figures#loc-preparing-figures-from-image-files. When you submit your revised manuscript, please ensure that your figures adhere fully to these guidelines and provide the original underlying images for all blot or gel data reported in your submission. See the following link for instructions on providing the original image data: https://journals.plos.org/plosone/s/figures#loc-original-images-for-blots-and-gels. 

Answer: Thanks for your kind suggestions. We have added the original uncropped and unadjusted images in a PDF and upload it for a supplement file.

In cover letter, we have noted that the blot/gel image data are in Supporting Information marked in red.

Question 6: Please include captions for your Supporting Information files at the end of your manuscript, and update any in-text citations to match accordingly. Please see our Supporting Information guidelines for more information: http://journals.plos.org/plosone/s/supporting-information.

Answer: Thanks for your kind suggestions. The original contributions presented in the study are all included in the article/Supplementary Material, further inquiries can be directed to the corresponding author.

Reviewer #1: 

Question 1: The authors described, “All procedures involving animal and experimental protocols were approved by the Hebei University of Chinese Medicine Animal Care and Use Committee.” (lines 99–102) without showing approval date and number(s).

Answer: Thanks for your kind suggestions. Thank you for reminding us that we have added the approval date and number in “Methods” marked in red. Thank you very much for your valuable advice.

Question 2: In Fig. 1, what age and gender of mice were used should be clarified. 

Answer: Thanks for your kind suggestions. We have added age and gender of mice in Fig. 1 according to your suggestion. Thank you very much for your valuable advice

Question 3: In Fig. 2, what age and gender of mice were used should be clarified.

Answer: Thanks for your kind suggestions. We have added age and gender of mice in Fig. 2 according to your suggestion. Thank you very much for your valuable advice

Question 4: In Fig. 3, what age and gender of mice were used should be clarified.

Answer: Thanks for your kind suggestions. We have added age and gender of mice in Fig. 3 according to your suggestion. Thank you very much for your valuable advice

Question 5: In Fig. 4, what age and gender of mice were used should be clarified.

Answer: Thanks for your kind suggestions. We have added age and gender of mice in Fig.4 according to your suggestion. Thank you very much for your valuable advice

Question 6: In Fig. 5, what age and gender of mice were used should be clarified.

Answer: Thanks for your kind suggestions. We have added age and gender of mice in Fig. 5 according to your suggestion. Thank you very much for your valuable advice

Question 7: In Fig. 6, what age and gender of mice were used should be clarified. In addition, scale bar(s) should be added in Fig. 6F.

Answer: Thanks for your kind suggestions. We have added age and gender of mice in Fig. 6 according to your suggestion. Thank you very much for your valuable advice. In addition, according to your suggestions, we have added scale bar(s) in New-Fig6.tif. 

Question 8: In Fig. 7, what age and gender of mice were used should be clarified.

Answer: Thanks for your kind suggestions. We have added age and gender of mice in Fig. 7 according to your suggestion. Thank you very much for your valuable advice

Question 9: In Fig. 8, what age and gender of mice were used should be clarified.

Answer: Thanks for your kind suggestions. The result shown in Fig 8 was carried out on H9C2 cell experiments. Thank you very much for your valuable advice. 

Question 10: In Suppl. Fig. 2, the arrow did not indicate any band/signal in Tubulin “55KD”. 

Answer: Thanks for your kind suggestions. According to your suggestions, we relabeled Tubulin “55KD” with arrows in Suppl. Fig. 2. Thank you very much for your valuable advice.

Question 11: In Suppl. Fig. 7, “dh/m”, “db/db”, and “CIH” did not match the band position. The 110 KD band in “Nuclear-Nrf2” did not fit in the figure.

Answer: Thanks for your kind suggestions. According to your suggestions, we have matched the “dh/m”, “db/db”, and “CIH” with the band position in Suppl. Fig.7. 

We have conducted WB results and replaced the representative image of “Nuclear-Nrf2” in Suppl. Fig. 7, as shown in New Fig. 7. Thank you very much for your valuable advice.

Question 12: In Suppl. Fig. 8, the 54 KD band(s) in “p-LKB1” were not clear. 

Answer: Thanks for your kind suggestions. We have conducted WB results and replaced the representative image of “p-LKB1” in Suppl. Fig. 8, as shown in New Fig. 8. Thank you very much for your valuable advice.

Question 13: “Dihydroethidium” (line 35) should be changed to “dihydroethidium”.

Answer: Thank you for your review and guidance. As recommended by reviewer, " Dihydroethidium " has been changed into " dihydroethidium" in line 35 marked in red.

Question 14: “Ismail Laher [9]” (line 66) should be changed to “Baran et al. [9]” or “Laher and his colleagues [9]”.

Answer: Thanks for your kind suggestion. As recommended by reviewer, " Ismail Laher [9]" has been changed into " Baran et al. [9]" in line 66 marked in red.

Question 15: “Hu k [10]” (line 69) should be changed to “Peng and Hu [10]”.

Answer: Thank you for your review and guidance. As recommended by reviewer, " Hu k [10]" has been changed into " Peng and Hu [10]" in line 69 marked in red.

Question 16: “37°C” (lines 153, 160, and 165) should be changed to “37 ˚C”.

Answer: Thanks for your kind suggestion. As recommended by reviewer, "37°C" has been changed into " 37 ˚C" marked in red.

Question 17: Concerning intermittent hypoxia in cardiomyocytes, several paperers (Hu, Z. et al. Sleep Breath doi: 10.1007/s11325-023-02863-8. Online ahead of print; Chen, Q. et al. Sleep Breath 27, 129-136, 2023; Dong, N. & Liu, W.Y. Sleep Breath 27, 399-410; Takasawa, S. et al. Int. J. Mol. Sci. 23, 8782, 2022; Takasawa, S. et al. Int. J. Mol. Sci. 23, 12414, 2022; Moulin, S. et al. Antioxidants 11, 1462, 2022) have recently been reported. The authors should cite the papers and make some discussions.

Question 18: There are some reports concerning human cardiomyocytes in intermittent hypoxia such as Regev, D. et al. Int. J. Mol. Sci. 23, 10272, 2022, Huang, J. et al. Sleep Breath 27, 1005-1011, 2023; Moulin, S. et al. Antioxidants 11, 1462, 2022; Chen, Q. et al. Sleep Breath 27, 129-136, 2023; Dong, N. & Liu, W.Y. Sleep Breath 27, 399-410). The authors should cite the papers and make some discussions.

Answer: Thanks for your kind suggestions. We have cited the papers and make some discussions marked in red in the manuscript according to your suggestions. The detailed modifications are as follows:

1.In the second paragraph of “Discussion”, we have cited the paper such as “Takasawa, S. et al. Int. J. Mol. Sci. 23, 8782, 2022”

2.In the third paragraph of “Discussion”, we have cited the paper such as “Regev, D. et al. Int. J. Mol. Sci. 23, 10272, 2022” and make some discussions marked in red.

3. In the fourth paragraph of “Discussion”, we have cited three papers such as Takasawa, S. et al. Int. J. Mol. Sci. 23, 12414, 2022; Hu, Z. et al. Sleep Breath; Dong, N. & Liu, W.Y. Sleep Breath and make some discussions marked in red.

4. In the sixth paragraph of “Discussion”, we have cited three papers such as Chen, Q. et al. Sleep Breath; Moulin, S. et al. Antioxidants 11, 1462, 2022; Huang, J. et al. Sleep Breath 27, 1005-1011, 2023 and make some discussions marked in red.

Reviewer #2: 

Question 1: Introduction: Page 11: “…and excessive ROS can inhibit AMPK inactivation by inhibiting the phosphorylation of LKB1”. Here, I think you meant ROS can inhibit AMPK activation, or?

Answer: Thank you very much for your valuable advice. We agree with your suggestion, and excessive ROS can inhibit AMPK activation. And we have consulted the relevant references [1-5]. The references are as follows:

[1] Zhang J, Zhu Y, Lai C, Du H, Tang K. Expression of sirtuin type 3 in locus ceruleus is associated with long-term intermittent hypoxia-induced neurocognitive impairment in mice. Neuroreport 2020;31(3):220-5.

[2] Wu Y, Duan X, Gao Z, Yang N, Xue F. AICAR attenuates postoperative abdominal adhesion formation by inhibiting oxidative stress and promoting mesothelial cell repair. PLoS One. 2022 Sep 1;17(9):e0272928. 

[3] Jiang P, Ren L, Zhi L, Yu Z, Lv F, Xu F, et al. Negative regulation of ampk signaling by high glucose via e3 ubiquitin ligase mg53. Mol Cell 2021;81(3):629-37. 

[4] Li Q, Tuo X, Li B, Deng Z, Qiu Y, Xie H. Semaglutide attenuates excessive exercise-induced myocardial injury through inhibiting oxidative stress and inflammation in rats. Life Sci. 2020 Jun 1;250:117531.

[5] Chen X, Li X, Zhang W, He J, Xu B, Lei B, et al. Activation of AMPK inhibits inflammatory response during hypoxia and reoxygenation through modulating JNK-mediated NF-κB pathway. Metabolism. 2018 Jun;83:256-270.

Question 2: Methods, page 11: only male animals were used. Was there any reason not to include females? Including only males is a limitation of this study. How old were the mice at the beginning of the experiment?

Answer: Thanks for your kind suggestions. We have used male mice in the study. Before using male mice, we consulted many related literatures and found that male mice were used in a large number of studies, and female mice were easily influenced by hormones, which complicated many factors [1-3]. In addition, combined with the clinical studies, we found that male OSA patients are at a higher risk of developing DM than female patients [4-5]. Finally, we chose male mice for study. In future studies, we will appropriately add female mice to observe the corresponding indicators. Thank you very much for your valuable advice.

These mice were 4 to 5 weeks old at the beginning of the experiment in the “Animals and treatment”of “Materials and methods” marked in red. Thank you very much for your valuable advice. The references are as follows:

[1] Badran M, Abuyassin B, Golbidi S, Ayas N, Laher I. Uncoupling of Vascular Nitric Oxide Synthase Caused by Intermittent Hypoxia. Oxid Med Cell Longev. 2016;2016:2354870. 

[2] Takahashi N, Yoshida H, Kimura H, Kamiyama K, Kurose T, Sugimoto H, Imura T, Yokoi S, Mikami D, Kasuno K, Kurosawa H, Hirayama Y, Naiki H, Hara M, Iwano M. Chronic hypoxia exacerbates diabetic glomerulosclerosis through mesangiolysis and podocyte injury in db/db mice. Nephrol Dial Transplant. 2020 Oct 1;35(10):1678-1688. 

[3] Takahashi N, Yoshida H, Kimura H, Kamiyama K, Kurose T, Sugimoto H, Imura T, Yokoi S, Mikami D, Kasuno K, Kurosawa H, Hirayama Y, Naiki H, Hara M, Iwano M. Chronic hypoxia exacerbates diabetic glomerulosclerosis through mesangiolysis and podocyte injury in db/db mice. Nephrol Dial Transplant. 2020 Oct 1;35(10):1678-1688. 

[4] Tao Y, Li X, Yang G, Wang L, Lian J, Chang Z. Gender Differences in the Association Between Obstructive Sleep Apnea and Diabetes. Diabetes Metab Syndr Obes. 2021 Nov 23;14:4589-4597. 

[5] Chung F, Liao P, Yang Y, Andrawes M, Kang W, Mokhlesi B, Shapiro CM. Postoperative sleep-disordered breathing in patients without preoperative sleep apnea. Anesth Analg. 2015 Jun;120(6):1214-24.

Question 3: Methods, page 11: “24 db/db mice were randomly divided into three groups (n=8): db/m group, db/db group, and CIH group”. Db/db mice cannot serve as a source of db/m mice. Please rewrite this sentence.

Answer: Thank you very much for your valuable advice. We have rewritten this sentence according to your suggestion in the second paragraph of “Animals and treatment”of “Materials and methods” marked in red. The detailed modifications are as follows: The db/db mice (n=16) were randomly divided into the db/db group and CIH group. The db/m mice were used as the control group (n=8).

Question 4: Methods, page 11: “The concentration of oxygen in the chamber for the CIH group was changed from 21% to 5%, 8 h/day for 8 weeks”. -> How long was one cycle of normoxia-hypoxia-normoxia? Did you measure the extent of hypoxemia in blood of mice under intermittent hypoxia?

Answer: Thank you very much for your valuable advice. One cycle of normoxia-hypoxia-normoxia was 5 minutes. In the first 3.5 minutes, 100% nitrogen was injected into the device to reduce the oxygen concentration to 5%, and in the next 1.5 minutes, oxygen was injected to gradually restore the oxygen concentration to 21%.

Thanks for your kind suggestions, we did not measure the extent of hypoxemia in the blood of mice with intermittent hypoxia. However, the oxygen concentration in the hypoxic chamber during modeling is guaranteed at 21% to 5%, and the model is stable, and the laboratory has also published relevant model articles [1-3]. We will be focused on the extent of hypoxemia in our future study. Thank you very much for your valuable advice. The references are as follows:

[1] Li D, Si J, Guo Y, Liu B, Chen X, Qi K, et.al. Danggui-Buxue decoction alleviated vascular senescence in mice exposed to chronic intermittent hypoxia through activating the Nrf2/HO-1 pathway. Pharm Biol. 2023 Dec;61(1):1041-1053. 

[2] Bingbing L, Jieru LI, Jianchao SI, Qi C, Shengchang Y, Ensheng JI. Ginsenoside Rb1 alleviates chronic intermittent hypoxia-induced diabetic cardiomyopathy in db/db mice by regulating the adenosine monophosphate-activated protein kinase/Nrf2/heme oxygenase-1 signaling pathway. J Tradit Chin Med. 2023 Oct;43(5):906-914. 

[3] Song JX, Zhao YS, Zhen YQ, Yang XY, Chen Q, An JR, et.al. Banxia-Houpu decoction diminishes iron toxicity damage in heart induced by chronic intermittent hypoxia. Pharm Biol. 2022 Dec;60(1):609-620.

Question 5: Methods, page 12: How did the authors measure glucose levels?

Answer: Thank you very much for your valuable advice. Blood glucose levels are measured every two weeks. The mice were fasted for 15 hours at 18:00 the night before each blood sugar measurement, and the blood glucose was measured at 9:00 the next morning.

Question 6: Methods, page 13: the authors describe paraffin embedment of heart specimens for subsequent histological stainings, but in the next method section, frozen sections are mentioned. The method of frozen section preparation is missing.

Answer: Thank you very much for your valuable advice. We have added the method of frozen section in the third paragraph of “Histological examination” of “Materials and methods” marked in red according to your suggestion. The detailed modifications are as follows: The heart was fixed on ice in 4% paraformaldehyde for 2 h, and then transferred to a 30% sucrose solution for dehydration at 4˚C overnight, embedded with OCT, sliced with a frozen microtome with a thickness at 5μm, and fixed in cold acetone for 10min.

Question 7: Methods, page 12: echocardiographic measurements are not precisely described: what view was used? What mode was used? E.g. M-mode for chamber diameters, B-mode for Doppler measurements.

Answer: Thank you very much for your valuable advice. We have precisely described the echocardiographic measurements in “Echocardiography” of “Materials and methods” marked in red according to your suggestion. The detailed modifications are as follows: B-mode is the basic imaging mode of ultrasonic imaging, in which images of the anatomical structure of animals are used to locate the long and short axes of mice. M-mode was used to measure the left ventricular end diastolic diameter (LVDd), left ventricular end systolic diameter (LVDs), left ventricular fractional shortening (LVFS), and left ventricular ejection fraction (LVEF). Color Doppler-mode was used to measure the velocity ratio of the E peak to the A peak in the cardiac mitral valve (E/A). 

Question 8: Methods, page 12: Did you measure thickness of the left ventricular walls and the potential impact of diabetes and IH on this parameter?

Answer: Thank you very much for your valuable advice. We have measured the thickness of the left ventricular end systolic/diastolic anterior wall thickness in the experiment. Compared to the db/m group, the LVAWs and LVAWd of the db/db group and the CIH group were increased (p < 0.01). Compared to the db/db group, the LVAWs and LVAWd of the CIH group were increased (p < 0.05). The results suggested that CIH increased thickness of the left ventricular walls in db/db mice

Question 9: Methods: The authors present results for CK-MB, troponin and LDH serum levels, the corresponding section in the Methods how these parameters were measured is missing.

Answer: Thank you very much for your valuable advice. We have added the method of the serum levels of CK-MB, LDH and cTnI in “The measurement of biochemical parameters” of “Materials and methods” marked in red according to your suggestion. The detailed modifications are as follows: The levels of creatine kinase myocardial band (CK-MB), lactate dehydrogenase (LDH) and cardiac troponin Ⅰ (cTnI) were detected by automatic blood biochemical detector.

Question 10: Methods, page 15: “we cultured the H9C2 cells with different concentrations of HG for 48 h.” -> what were the concentrations of HG?

Answer: Thank you very much for your valuable advice. We have added the concentrations of HG in “Cell culture and treatment” of “Materials and methods” marked in red according to your suggestion. The detailed modifications are as follows: We cultured the H9C2 cells with 30mM of HG for 48 h.

Question 11: Methods, page 17: Statistical analysis: did the authors check for normal distribution? What method was used for the comparisons between two groups?

Answer: Thanks for your kind suggestions. The normal distribution has been checked. One-way ANOVA was used for data comparison between groups, and LSD was used for statistical analysis when the variance of pairwise comparison was homogeneous. Dunnett's T3 was used for statistical analysis of variance heterogeneity. The P<0.05 was statistically significant. Thank you very much for your valuable advice.

Question 12: Results, page 23: the results showed that metformin improved membrane potential and decreased oxidative stress in cells which was associated with increased p-AMPK/AMPK and Nrf2 levels. Did this have any impact on the cell viability/apoptosis?

Answer: Thanks for your kind suggestion. Metformin can improve the cell viability and reduce apoptosis. We have provided data on the effect of metformin on cell viability and apoptosis. And we found that metformin improved the cell viability and alleviated apoptosis. Thank you very much for your valuable advice.

 Question 13: Graphical abstract: according to the findings, not only cardiomyocyte apoptosis, but also aggravated fibrosis may lead to cardiac dysfunction; fibrosis is not illustrated here.

Answer: Thanks for your kind suggestion. We have revised the Graphical abstract and fibrosis has been illustrated. The detailed modifications are as follows:

Question 14: Throughout the whole manuscript there are sentences which can hardly be understood (below only a couple of examples of them). Please improve this as it might be very confusing for the reader.

Answer: Thank you for your review and guidance. Our manuscript has been send to native English speaking editors for improving writing style and checking the whole manuscript as showed below. 

Results, page 17: “Compared to the db/db group, the BW of the CIH group was decreased (p<0.05), but there was no statistical significance in FBG and the fluctuation of blood glucose in the CIH group was increased.” -> This sentence is unclear.

Answer: Thank you for your review and guidance. We have rewritten this sentence marked in red according to your suggestion. Our manuscript has been send to native English speaking editors for improving writing style and checking the whole manuscript. The detailed modifications are as follows: Compared to the db/db group, the BW of the CIH group was decreased (p < 0.05). There was no statistical significance in FBG between the db/db group and the CIH group, but the fluctuation of FBG in the CIH group was increased.

Results, page 19: “As shown in Fig.3A, the results showed that myocardial disorders in the db/db group and the CIH group compared to the db/m group.” -> This sentence is unclear.

Answer: Thank you for your review and guidance. We have rewritten this sentence marked in red according to your suggestion. Our manuscript has been send to native English speaking editors for improving writing style and checking the whole manuscript. The detailed modifications are as follows: As shown in Fig.3A, compared to the db/m group, the results showed that myocardial disorders in the db/db group and the CIH group.

Figure 8, legend: “The cell viability of H9C2 cells which was treated with IH and HG for 24 h and 48h.” -> This sentence is unclear.

Answer: Thank you for your review and guidance. We have rewritten this sentence marked in red according to your suggestion. Our manuscript has been send to native English speaking editors for improving writing style and checking the whole manuscript. The detailed modifications are as follows: The H9C2 cells were treated with IH and HG for 24 and 48 h, and cell viability of H9C2 cells was measured.

---

## [Decision Letter · Decision Letter 1]

19 Dec 2023

Chronic intermittent hypoxia aggravated diabetic cardiomyopathy through LKB1/AMPK/Nrf2 signaling pathway

PONE-D-23-28549R1

Dear Dr. Yang,

We’re pleased to inform you that your manuscript has been judged scientifically suitable for publication and will be formally accepted for publication once it meets all outstanding technical requirements.

Kind regards,

Michael Bader

Academic Editor

PLOS ONE

Additional Editor Comments (optional):

Reviewers' comments:

Reviewer's Responses to Questions

**Comments to the Author**

1. If the authors have adequately addressed your comments raised in a previous round of review and you feel that this manuscript is now acceptable for publication, you may indicate that here to bypass the “Comments to the Author” section, enter your conflict of interest statement in the “Confidential to Editor” section, and submit your "Accept" recommendation.

Reviewer #1: All comments have been addressed

Reviewer #2: All comments have been addressed

2. Is the manuscript technically sound, and do the data support the conclusions?

Reviewer #1: Yes

Reviewer #2: Yes

3. Has the statistical analysis been performed appropriately and rigorously? 

Reviewer #1: Yes

Reviewer #2: Yes

4. Have the authors made all data underlying the findings in their manuscript fully available?

Reviewer #1: Yes

Reviewer #2: Yes

5. Is the manuscript presented in an intelligible fashion and written in standard English?

Reviewer #1: Yes

Reviewer #2: Yes

6. Review Comments to the Author

Reviewer #1: All the points that I pointed out were suitably revised in PONE-D-23-28549R1. It is a very nice paper. Congratulations!

Reviewer #2: Dear Authors,

thank you very much for all answers and corresponding changes in the revised manuscript.

7. PLOS authors have the option to publish the peer review history of their article (what does this mean?). If published, this will include your full peer review and any attached files.

Reviewer #1: No

Reviewer #2: No

---

## [Editor Report · Acceptance letter]

28 Feb 2024

PONE-D-23-28549R1 

PLOS ONE

Dear Dr. Yang, 

I'm pleased to inform you that your manuscript has been deemed suitable for publication in PLOS ONE. Congratulations! Your manuscript is now being handed over to our production team.

Kind regards, 

on behalf of

Prof. Michael Bader 

Academic Editor

PLOS ONE